# Integrated hydrodynamic and machine learning models for compound flooding prediction in a data-scarce estuarine delta

Joko Sampurno[1,2], Valentin Vallaeys[1], Randy Ardianto[3], Emmanuel Hanert[1,4]

[1]Earth and Life Institute (ELI), Université Catholique de Louvain (UCLouvain), Louvain-la-Neuve, 1348, Belgium
5  [2]Department of Physics, Fakultas MIPA, Universitas Tanjungpura, Pontianak, 78124, Indonesia
[3]Pontianak Maritime Meteorological Station, Pontianak, 78111, Indonesia
[4]Institute of Mechanics, Materials and Civil Engineering (IMMC), Université Catholique de Louvain (UCLouvain), Louvain-la-Neuve, 1348, Belgium

*Correspondence to*: Joko Sampurno (joko.sampurno@uclouvain.be, jokosampurno@physics.untan.ac.id)

10  **Abstract.** Flood forecasting based on hydrodynamic modeling is an essential non-structural measure against compound flooding over the globe. With the risk increasing under climate change, all coastal areas are now in need of flood risk management strategies. Unfortunately, for local water management agencies in developing countries, building such a model is challenging due to the limited computational resources and the scarcity of observational data. We attempt to solve this issue by proposing an integrated hydrodynamic and machine learning approach to predict water level dynamics as a proxy of 15 compound flooding risk in a data-scarce delta. As a case study, this integrated approach is implemented in Pontianak, the densest coastal urban area over the Kapuas River delta, Indonesia. Firstly, we built a hydrodynamic model to simulate several compound flooding scenarios. The outputs are then used to train the machine learning (ML) model. To obtain a robust machine learning model, we consider three machine learning algorithms, i.e., Random Forest, Multi Linear Regression, and Support Vector Machine. Our results show that the integrated scheme works well. The Random Forest (RF) is the most accurate 20 algorithm to model water level dynamics in the study area. Meanwhile, the machine-learning model with the RF algorithm can predict eleven out of seventeen compound flooding events during the implementation phase. It could be concluded that RF is the most appropriate algorithm to build a reliable ML model capable of estimating the river water level dynamics within Pontianak, whose output can be used as a proxy for predicting compound flooding events in the city.

## 1 Introduction

25  Compound flooding in low-lying coastal areas is a recognized hazard that can be exacerbated by global warming (Hao and Singh, 2020; Santiago-Collazo et al., 2021; Gori et al., 2022; Hsiao et al., 2021; Ghanbari et al., 2021). Compound flooding hazard is derived from the interaction of storm surge penetration, riverine flooding, and intense rainfall over the areas (as the impact of extreme meteorological events) that coincide or nearly coincide (Bilskie and Hagen, 2018; Ikeuchi et al., 2017; Wahl et al., 2015). This natural hazard can endanger the population and the coastal area's infrastructures, which have been growing 30  fast in the last decade (Bhaskaran et al., 2014). Without appropriate mitigation, the consequences of the hazard can be severe

for the coastal environment (Costabile et al., 2013) and the coastal communities both economically (Karamouz et al., 2014) and socially (Comer et al., 2017).

There are various mechanisms driving compound flooding in low-lying urban coastal areas (Santiago-Collazo et al., 2019). First, the water level increases with the tide, and the sea level rises due to climate change. On top of this, a storm surge may occur. The water can get into the dry land by wave overtopping. Second, extreme precipitation and a high upstream flow discharge can also elevate water in a low-lying delta. In this case, water can overflow and cause flooding as well. These flood pathways are often naturally correlated, so those mechanisms occur coincidentally (or in close succession), creating a compound event and worsening the hazard.

Flood forecasting based on water-level prediction in a tidal river area is an essential non-structural measure against compound flooding (Chan, 2015; Tucci and Villanueva, 1999; Mosavi et al., 2018). Non-structural measures mean any actions to manage the risk of compound flooding without involving a physical construction (UNDRR, 2022), including land-use regulations, flood forecasting, warning systems, floodproofing and disaster prevention, and preparedness and response mechanisms. The water level could be predicted using a process-based or data-based approach. The process-based approach is more commonly used to tackle the water-level prediction issue (Costabile and Macchione, 2015; Ye et al., 2021), but it requires many assumptions to reduce the complexity—making it computationally tractable. The data-based approach, e.g., machine learning and statistical models, can also predict water level changes and compound flooding without the underlying physical attributes and high computational resources (Choi et al., 2020; Wang and Wang, 2020; Assem et al., 2017; Couasnon et al., 2020; Bevacqua et al., 2019). Machine learning involves developing a model that can improve task performance over time by learning from examples, with minimal human efforts instructing them how to do so. Machine learning allows users to test hypotheses and generate confidence bonds for mitigation strategies. Machine learning models can capture and represent a complex input and output relationship using only historical data (Chen and Asch, 2017). For instance, by assuming that flood events are stochastic, machine learning can predict major flood events based on certain probability distributions from the historical discharge data (Mosavi et al., 2018). In some cases, their performance is even more accurate than traditional statistical models (Xu and Li, 2002). In other words, we can prepare strategies to mitigate the flood risks using a machine learning model.

However, building a flood forecasting model in developing countries can be challenging. Implementing a process-based approach requires expensive computational resources (Nayak et al., 2005). Meanwhile, resources owned by local agencies are often limited, so local operational management may not have access to it. Additionally, building a robust machine learning model requires a sufficient amount of data for the training (Naqa et al., 2018), but the availability of observational data in these areas is also limited. Some studies proposed remote sensing techniques (optical and SAR images) as a solution (Mokkenstorm et al., 2021; Kabenge et al., 2017; Haq et al., 2012). Nevertheless, due to the limitation of its time resolution, the technique cannot always detect compound flooding. Therefore, a remote sensing technique is more suitable for detection, monitoring, validation, and mitigation purposes instead of for prediction.

A new paradigm that combines deterministic and machine learning components has been proposed to tackle data and computational limitations in environmental modeling, such as hybrid climate models (Krasnopolsky and Fox-Rabinovitz,

2006) and an ML model for 2D surface water catchment problems (Maxwell et al., 2021). However, to the best of our knowledge, no previous modeling frameworks have developed a deterministic model to train a machine learning model for compound flooding studies. As a common practice, compound flood modeling typically uses the coupling of two or more hydrodynamic, hydraulic, or hydrological models (Hsiao et al., 2021; Santiago-Collazo et al., 2021; Ikeuchi et al., 2017). The coupling could be one-way, two-way, or dynamic coupling. Another approach is deep learning and data fusion (Muñoz et al.,

2021), and data assimilation (Muñoz et al., 2022).

This study attempts to fill the gap by combining the process-based and data-based approaches as a state-of-the-art framework to predict water level dynamics, a proxy for compound flooding in a data-scarce delta. Firstly, we build a hydrodynamic model to run some flood scenarios in a data-scarce estuary. Then, we create machine learning models trained using the hydrodynamic model's outputs to predict the water level and forecast future floods. To obtain a robust machine learning model, we evaluate

three machine learning algorithms and select the most accurate one for our application. As a case study, the integrated framework is implemented in the city of Pontianak, whose population density is the highest within the Kapuas River delta. This city experienced a compound flooding event on 29 December 2018 (Sampurno et al., 2022), and the impact was severe (Madrosid, 2018). At that moment, the water level dynamic is about to go down after passing its peak elevation, when suddenly a strong force pushes it to go up again for a short moment. The interaction between tides, storm surges, and discharges along

the tidal river in the Kapuas River delta is responsible for a 30 cm increase in the water level during the event. The finding is expected to assist the local water management agency in assessing their compound flood hazards and mitigating their risk despite the limited data and computational resources.

## 2 Material and method

### 2.1 Study area

The Kapuas River is the longest inland river in Indonesia (Goltenboth et al., 2006). The basin is located in the western part of Borneo Island (Fig. 1). The water catchment area spreads over about 93,000 km$^2$ (about 12.5% of the Borneo Island area, Fig. 1), with about 66.7% of it consisting of forests (Wahyu et al., 2010). The upstream topography comprises hills covered mainly by Acrisol soils (Fig. 2), and the downstream consists of plains with more heterogeneous soil types (Fig. 2), such as *Humic Gleysols* (derived from grass or forest vegetation) and *Dystric Fluvisols* (young soil in alluvial deposits). The river is vital for

the local communities as a source of fresh water and a transportation system.

In the last decades, palm oil cultivation and forest fires expanded massively in the Kapuas water catchment (Semedi, 2014; Jadmiko et al., 2017). These circumstances changed the Kapuas hydrological regime and triggered more intense flooding in the river's floodplains. Combined with global sea-level rise, these phenomena could lead to more intense and severe flood events, particularly in the river delta.

The delta of the Kapuas River is still mostly natural, with no dams, dykes, or groins on its downstream. Therefore, the hydrodynamics of the river significantly influences the flood occurrences in the delta. The most populated area over the delta is Pontianak, a city located in the Kapuas Kecil—the middle stream of the second-largest branch of the Kapuas River.

As a tidal river, the tidal regime within the Kapuas River delta is mixed but mainly diurnal (Kästner, 2019). The dominant tidal constituent is K1, O1, P1, M2, and S2 (Pauta, 2018). The average tidal amplitude within the delta is set in a microtidal

regime, with a mean spring range of 1.45 m at its river mouth (Kästner, 2019).

## 2.2 Hydrodynamic model description

To simulate hydrodynamics within the Kapuas River delta, we use the multi-scale hydrodynamic model SLIM 2D (Lambrechts et al., 2008; Gourgue et al., 2009; Remacle and Lambrechts, 2016). The model can simulate hydrodynamic processes along the land-sea continuum, from the river to the ocean (Vallaeys et al., 2018, 2021; Frys et al., 2020; Le et al.,

2020b). We simulate compound flooding events based on the water level dynamics for different forcing scenarios using the model. The model solves the 2D Shallow Water Equations (SWE):

$$\frac{\partial H}{\partial t} + \nabla \cdot U = R, \tag{1}$$

$$\frac{\partial U}{\partial t} + \nabla \cdot \left(\frac{UU}{H}\right) + f\boldsymbol{e_z} \times U = \nabla \cdot (\upsilon \nabla U) + \alpha g H \nabla (H - h) - \frac{Cd}{H^2}|U|U + \frac{1}{\rho}\tau_{wind} - \frac{H}{\rho}\nabla P_{atm} \tag{2}$$

where $H$ is the water column height, $\nabla$ is the horizontal gradient operator, $U = H\overline{u}$ is the horizontal transport, R is the rainfall,

$t$ is the time, $\overline{u} = (u, v)$ is the depth-averaged horizontal velocity, f is the Coriolis parameter, $\boldsymbol{e_z}$ is the vertical unit vector pointing upward, $\upsilon$ is the horizontal eddy viscosity, $\alpha$ is a constant to define a dry elements ($\alpha = 0$) and wet elements ($\alpha = 1$) (Le et al., 2020a), $h$ is the bathymetry, $g = 9.81$ m/s² is the gravitational acceleration, $Cd$ is the bulk drag coefficient, $\tau_{wind}$ is the wind stress and $\nabla P_{atm}$ is the atmospheric pressure gradient.

## 2.3. Metrics for model performance evaluation

We use the Nash–Sutcliffe efficiency (NSE) measure to evaluate the models' performance. NSE is used to assess the performance of the machine learning models in producing the predicted water level. A perfect model corresponds to NSE = 1, while a model that has the same predictive skill as the mean of the observed data represents by NSE = 0. Meanwhile, NSE<0 implies that the mean value of observed data predicts better than the model. The closer the NSE value to 1, the better the predictive skill of the model. The NSE coefficient is calculated as follows:

$$NSE = 1 - \frac{\sum_{t=1}^{T} \{H_m^t - H_0^t\}^2}{\sum_{t=1}^{T} (H_0^t - \overline{H_0})^2} \tag{4}$$

where $H_m^t$ represents the water level model at time $t$, $H_0^t$ represents the observed water level at the same time, and $\overline{H_0}$ is the mean of the observed water level.

Root Mean Square Errors (RMSE) of peaks between predicted water level and observation during the flood events are also used as an additional performance indicator. RMSE is used to represent the model's ability to predict flood events. The RMSE between the model outputs and the observations is calculated by:

$$RMSE = \sqrt{\frac{\sum_{i=1}^{n}(x_i - y_i)^2}{N}} \tag{5}$$

where $x_i$ is the water level as the model's output at the $i$-$th$ peaks, and $y_i$ is the observed water level at the same time. $N$ is the number of the total peak data.

## 2.4 Hydrodynamic model setup and calibration

In order to run the hydrodynamic model, we defined a computational domain that covers both the river and the ocean parts. Next, we generated an unstructured mesh to cover the domain, with a resolution of 50 m over the riverbanks, 400 m over the coast near the river mouth, 1 km over the rest of the coastline, and 5 km over the offshore (Fig. 3). The multi-scale mesh was generated using an algorithm developed by Remacle and Lambrechts (2018). Next, we set the bathymetry constructed from two data sets: first, the river and estuary bathymetry maps, obtained from the Indonesian Navy (Kästner, 2019), and second, the Karimata Strait bathymetry, obtained from BATNAS (BATimetri NASional, 2021). Furthermore, we set the bulk bottom drag coefficients, which are $2.5 \times 10^{-3}$ over the ocean (which corresponds to a sandy seabed) and $1.9 \times 10^{-2}$ over the river bed (Kästner et al., 2018). Lastly, we imposed the rainfall, as observed by the Pontianak Maritime Meteorological Station (PMMS).

The hydrodynamic model simulation is forced by wind and atmospheric pressure from ECMWF (Hersbach et al., 2020), and tides from TPXO (Egbert and Erofeeva, 2002). As upstream boundary conditions, we imposed discharge from the Kapuas River and the Landak River. The discharge data were retrieved from the Global Flood Monitoring System (GFMS) (Wu et al., 2014) at about 70 km and 40 km from the river mouth (Fig. 4). Since the GFMS calculates the flow using Integrated Multi-Satellite Retrievals for GPM (IMERG) precipitation information as input, the coastal processes do not affect the model output (predicted river flow).

We also imposed runoff, obtained by converting rainfall over the Kapuas Kecil River catchment area as an inlet water flux at 15 channels entering the domain (Fig. 4). The runoff of every channel was calculated from rainfall data using SWAT+ (Bieger et al., 2017), which considered the pressure, the humidity, and other weather parameter input. Here, we use one-way coupling, where the SWAT+ model runs first and independently. The SWAT+ model only produces the flow of channels that enter the river stream within the KRD. Then, we used these channel outlets as boundary conditions for the SLIM model. Unfortunately, during the tuning of the SWAT+ model, the correlation between the model's output (runoff) and the observation data is still low (Pearson correlation coefficient = 0.32). However, we decided to use the output as the channels' inlet boundary condition in the hydrodynamic model because the channel runoff volume is much less than the river discharge. Therefore, we assumed that it does not significantly affect the hydrodynamics of the river.

To evaluate the SLIM model performance, we ran a simulation for January 2019 and compared the simulated water elevation with the observations in Pontianak. The model errors correspond to an NSE of 0.87 and an RMSE of 0.12 m (Fig. 5). This RMSE is deemed sufficiently small to consider model outputs as a good proxy of the real system (Moriasi et al., 2015). We simulated the hydrodynamics with different oceanic, atmospheric, and river forcings to forecast flood events based on the water levels in Pontianak. Based on the Pontianak Maritime Meteorological Station report, the city is flooded when the water level exceeds 2.5 m. We, therefore, set this value as the threshold of a flood event. We ran the hydrodynamic model for ten months and extracted the output hourly to produce the scenarios (see Table 1). Then, we selected 6,000 sample points of the predicted water levels at Pontianak with their associated input dataset using a random sampling technique. We merged the data as a single dataset to train the machine learning model, encompassing all possible flood events resulting from the combination of the external forcings. The dataset shows that several flooding occurred within the simulations, indicated by sample points with water elevations greater than 2.5 m (Fig. 6).

## 2.5 Machine learning model

### 2.5.1 Dependent and predictor variables

To develop the machine learning models, we used the river water level at Pontianak as the dependent variable. Then, we considered atmospheric, oceanic, and riverine variables as predictors of the water level in the city. Atmospheric variables include average and maximum wind speed, wind direction, precipitation, and average atmospheric pressure. Oceanic variables cover tides at the river mouth, and the riverine variables consist of the Kapuas River, and the Landak River discharges. To evaluate the impact of each predictor before the flood event, we imposed the prior state (one and two hours before) of these parameters (see Table 2). The datasets were recorded hourly and combined with the SLIM output (used in the training and testing phases) and the observational data (used in the implementation phase).

Mutual Information ($MI$), a statistic tool that can measure the degree of relatedness between variables in a dataset, was implemented to evaluate the relation between each predictor and the dependent variable (Fig. 7). The greater the $MI$ value between two variables, the stronger the relatedness, regardless of how nonlinear its dependency is (Kinney and Atwal, 2014). $MI$ between two variables ($X$ and $Y$) is obtained from Choi et al. (2020):

$$MI = \sum_{x \in X} \sum_{y \in Y} p(x,y) \, log\left(\frac{p(x,y)}{p(x)-p(y)}\right) \tag{3}$$

where $p(x,y)$ is the joint probability distribution.

All predictors considered in the machine learning model have an $MI$ coefficient greater than zero, which means all predictor variables impact the river water level in Pontianak (Fig. 7). The relationship between these predictors and the water level could be linear or non-linear (as shown by the MI capturing both relation types). Here, we found that the tidal elevations in the river mouth (X1, X2, and X3) have the most decisive impact on the river water level in the city ($MI > 0.5$), while tidal elevation is observed one hour before (X2) is the strongest one. Next, the wind speed (max and average), the discharges (from both the

Kapuas and the Landak river), and the pressure have a moderate relatedness. In contrast, the wind direction and the rainfall have only a weak relatedness ($MI < 0.1$). This means that both parameters have no significant impact.

### 2.5.2 Machine learning algorithm

Here, we consider three different machine learning algorithms, i.e., random forest (RF), multiple linear regression (MLR), and support vector machine (SVM). RF is a supervised learning algorithm that operates by constructing many decision trees during the training (Breiman, 2001). The algorithm can be implemented for classification or regression. The model aggregates its multiple decision tree outcomes to generate the ultimate output, which is called the sub-sample outcomes (Han et al., 2012). The technique was enhanced by combining bootstrap in its aggregating processes (Breiman, 2001). Using this strategy, the

algorithm became an effective tool for classification and regression. In this study, the RF algorithm was obtained from the R randomForest library (Liaw and Wiener, 2002). To obtain the optimal parameter for the RF, we first tune the algorithm by searching for the optimal value of the number of variables randomly sampled as candidates at each split (mtry). As a result, the optimal number is 16 (Fig. 8).

MLR is a statistical technique that uses several explanatory variables to predict the outcome of a response variable (James et

al., 2013). This method fits the linear relationship between input features and the target (observed data) using the least-squared approach. In the least-squared approach, the best relationship model will be obtained by minimizing the sum of the squared distance between the calculated values (as model outputs) and the target values (James et al., 2013). This algorithm is the most straightforward approach in machine learning models and is generally used as the baseline method. The MLR algorithm implemented in this study was obtained from the R RWeka library (Hornik et al., 2008).

To obtain the best performance of the MLR algorithm, we did a statistical analysis to evaluate the multicollinearity among the predictor variables using the Variance Inflation Ratio (VIF). Since multicollinearity negatively affects the performance of the MLR model, VIF can help reduce the number of predictors (Alipour et al., 2020). Here, we found that some variables have VIF more significant than 5, which indicates a potentially severe correlation between these variables in the model (Fig. 9). Therefore, combined with the output of $MI$ analysis, we removed some variables which have low $MI$ and high VIF.

SVM is a supervised machine learning algorithm based on statistical learning frameworks (Gholami and Fakhari, 2017). This method is robust for modeling a complex non-linear relationship. The kernel function transforms the input features into a high-dimensional space to tackle the complexity. This transforms the non-linear relationship of input features into linear ones. Finally, linear regression is carried out to obtain the ultimate output. Compared with the other algorithms, SVM needs less computational resources because it can be trained only by a few features (Gholami and Fakhari, 2017). Previously, SVM was

only implemented for classification purposes, but it has also been implemented for regression purposes after some enhancement. The SVM algorithm implemented in this study was obtained from the R MARSSVRhybrid library (MARSSVRhybrid: MARS SVR Hybrid, 2021).

Since kernel function is critical in SVM, we tuned the SVM algorithm to obtain good results by selecting the most appropriate kernel parameter. We tested four kernels, i.e., linear, polynomial, radial basis, and sigmoid, as the candidates. We found that

the radial basis kernel performed the best for the SVM algorithm.

### 2.6 Model limitations

During the development process, we encountered potential errors that could be highlighted as model limitations. Firstly, we assumed that channel runoff volume would not affect the hydrodynamics of the river due to its small volume compared to the riverine volume. The average daily discharge of the Kapuas River and the Landak River during the simulation is about 4,137

$m^3/s$ and 406 $m^3/s$. At the same time, the total daily runoff of all channels which enter the hydrodynamic model domain in the KRD is about 32 $m^3/s$. The runoff contributes only about 0.7% of the total inlets in the hydrodynamic simulations; therefore, we assumed it is insignificant.

Secondly, we assumed that all the possible compound flood scenarios would occur within ten months. Since we already set some extreme values in the predictor parameters during the time, we assumed that all possible causes that drive compound

flooding in the domain are represented. However, this assumption may not be accurate.

Next, we only imposed the runoffs as inlets on the river banks in the hydrodynamic model domain. Hence, the model did not capture the hydrodynamic processes in the channels within the city. It means that the inundation processes in Pontianak were still not well represented. The model still lacks drainage systems for the urban region.

Moreover, the accuracy of the machine learning model depends on the hydrodynamic model's accuracy. The more accurate

the hydrodynamic model in predicting observational floods, the better the machine learning model will perform. Therefore, we need to tune the hydrodynamic model as accurately as possible.

Furthermore, since the rainfall impact on river water level is minor compared to other parameters, the model could not optimally capture urban flooding due to excessive rainfall. Based on the field observation, the city is shortly inundated if rain falls excessively for a few hours. This inundation could be due to the poor quality of the urban drainage system. Unfortunately,

this phenomenon is not directly captured by the water level observation located within the river. The increase in the river water level due to the heavy rain is minor.

Lastly, the model relies on the predicted input parameters such as weather parameters and river discharges to predict the future water level. Consequently, the more biased the predictors, the higher the uncertainty in the water-level prediction. Therefore, observational data as input parameters are needed to reduce the uncertainty and create a more robust model.

### 245   3 Results

During the training and testing phases, all NSE coefficients are greater than 0.8 both in the training and testing phases, which means that all algorithms perform very well. The most accurate algorithm is RF, followed by SVM and MLR (Fig. 10). As such, we know that all the tested machine learning algorithms are promising and need to be evaluated in the implementation phase using observational data.

Therefore, we implemented the machine learning models on the selected observational data, which were obtained during the high discharge season for three months in three years when inundations occurred (Dec 2018, Jan 2020, Jan 2021). Fig. 11 shows each proposed algorithm's predicted water levels compared to the observational data. Subsequently, the accuracy of models to predict flooding events, marked by points in Fig. 11, is evaluated.

Even though all algorithms performed very well during the training and testing phases, the performances differed during the

implementation phase (Table 3). However, the RF showed high accuracy in three different implementation phases. From the three different observational datasets, RF's NSE values range from 0.61 to 0.72, which is a good performance.

While the MLR algorithm succeeded in the training and testing phases, it only succeeded in the first and third implementation phases, with NSE of 0.72 and 0.65, respectively. The model was less successful in the second implementation phase, with NSE hitting only 0.35 for this implementation dataset.

Next, the SVM algorithm's performance is similar to the MLR algorithm. It succeeded in the training and testing phases but only succeeded in the first and third implementation phases, with NSE reaching 0.71 and 0.63, respectively. However, it failed in the second implementation dataset, with an NSE of only 0.41, which is slightly better than MLR.

Regarding flood events prediction, the RF algorithm also performed better than the other algorithms. It could predict eleven out of seventeen events (65% accuracy). On the other hand, MLR and SVM could only predict six and ten events (35% and

59% accuracy, respectively). Therefore, we know that the RF is the most accurate machine-learning algorithm to predict floods for our test case.

Unfortunately, these three algorithms also predicted false-positive events, i.e., flood events that never occurred during implementation (Table 3). While the RF predicted four false events, the MLR and the SVM predicted three false events. This false event prediction is the shortcoming of the algorithm, which should be addressed in future studies.

**4 Discussion**

The two main issues that have been tackled in this study are data scarcity and low computational resources for building flood forecasting models based on the water level dynamics in developing countries (Brocca et al., 2020; Singh et al., 2021). Here, we showed that using an approach that combines hydrodynamic and ML models is promising for obtaining a reliable and robust water level model. We succeeded in building and evaluating ML models trained by the hydrodynamic model output;

hence, they did not require extensive observational data in their training phase and did not need high computational costs in their implementation. Therefore, the proposed model is reliable for areas where observational data are scarce and computational resources are limited.

Since the proposed model can accurately forecast water levels, local water management agencies can rely on the model outputs for flood forecasting. Since machine learning does not require high computational resources, limited computational resources

will not hinder the assessment and mitigation of compound flooding hazards. Using the model, agencies can re-assess their

compound flood hazards and predict future events. Moreover, once they have more observation data, they can use it to re-adjust the proposed model or build a more robust one (Muñoz et al., 2021).

Next, we found that the RF algorithm is the best ML algorithm to predict water level as a proxy for compound flooding in the area of interest. In general, the performances of all tested ML algorithms for water level prediction are reasonable and

acceptable. However, considering the NSE values in all implementation phases, the number of flood events that are accurately predicted, and how close the predicted water level is during the events, it could be concluded that the RF performs better than other algorithms. The superiority of the RF algorithm in predicting water levels has also been shown in previous studies in the Upo Wetland (Choi et al., 2020) and the Poyang Lake (Li et al., 2016). Therefore, we proposed a machine learning model with the RF algorithm as the most appropriate model for the study area.

In addition, we found that the tidal elevation measured one-hour prior at the river mouth is the main parameter controlling the river water level in Pontianak. Even though the city is located 20 km from the river mouth, the tidal dynamics still strongly affect the river water level in the city. This result confirms previous studies, revealing that the tide propagation on the Kapuas River dominantly controls the river water level up to 30 km upstream (BGD - Modeling interactions between tides, storm surges, and river discharges in the Kapuas River delta, 2021), and still impacts up to 285 km from the river mouth (Kästner et

al., 2019).

Overall, our integrated approach can provide a model to predict compound flooding driven by the interaction of tide, wind surge from the oceanward, and high discharge from the river upstream. Regarding the limitation of the chosen indicator's capability to capture flood events, we will look for more data and indicators to enhance the model capability in future studies. Moreover, we will reduce the number of predictors to minimize the model output's uncertainty. We will also evaluate mean

sea-level rise due to climate change to broaden the model implementation and create better flood mitigation.

**5 Conclusion**

This study shows that an integrated approach between the hydrodynamic and the machine learning models successfully overcomes modeling river water-level and predicting compound flooding hazards in a data-scarce environment with limited computational resources. Therefore, the approach is suitable for local water management agencies in developing countries that

are faced with these issues. However, the accuracy of the machine learning model depends on the accuracy of the hydrodynamic model. If the hydrodynamic model is inaccurate in predicting real-life floods, the machine learning model's accuracy will also be lower. Besides, it has not yet optimally captured the urban flooding due to excessive rainfall. Considering more indicators representing this kind of flooding is essential to enhance the model's capability in the future. Regarding the implementation in Pontianak, we found that the machine learning model with the RF algorithm has the most accurate output

compared to the other algorithms. In addition, the tidal elevation, measured one hour prior, is the main predictor for water level modeling in the study area.

**Author contribution**

JS, VV, and EH conceptualized the research; JS and RA curated the data; JS, VV, and EH analyzed the data; JS wrote the manuscript draft; JS and EH reviewed and edited the manuscript.

**Competing interests**

The authors declare that they have no conflict of interest.

**Acknowledgment**

The PhD fellowship of Joko Sampurno is provided by Indonesia Endowment Fund for Education (LPDP) under Grant No. 201712220212183. Computational resources have been provided by the supercomputing facilities of the Université catholique
de Louvain (CISM/UCL) and the Consortium des Équipements de Calcul Intensif en Fédération Wallonie Bruxelles (CÉCI) funded by the Fond de la Recherche Scientifique de Belgique (F.R.S.-FNRS) under convention 2.5020.11 and by the Walloon Region.

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

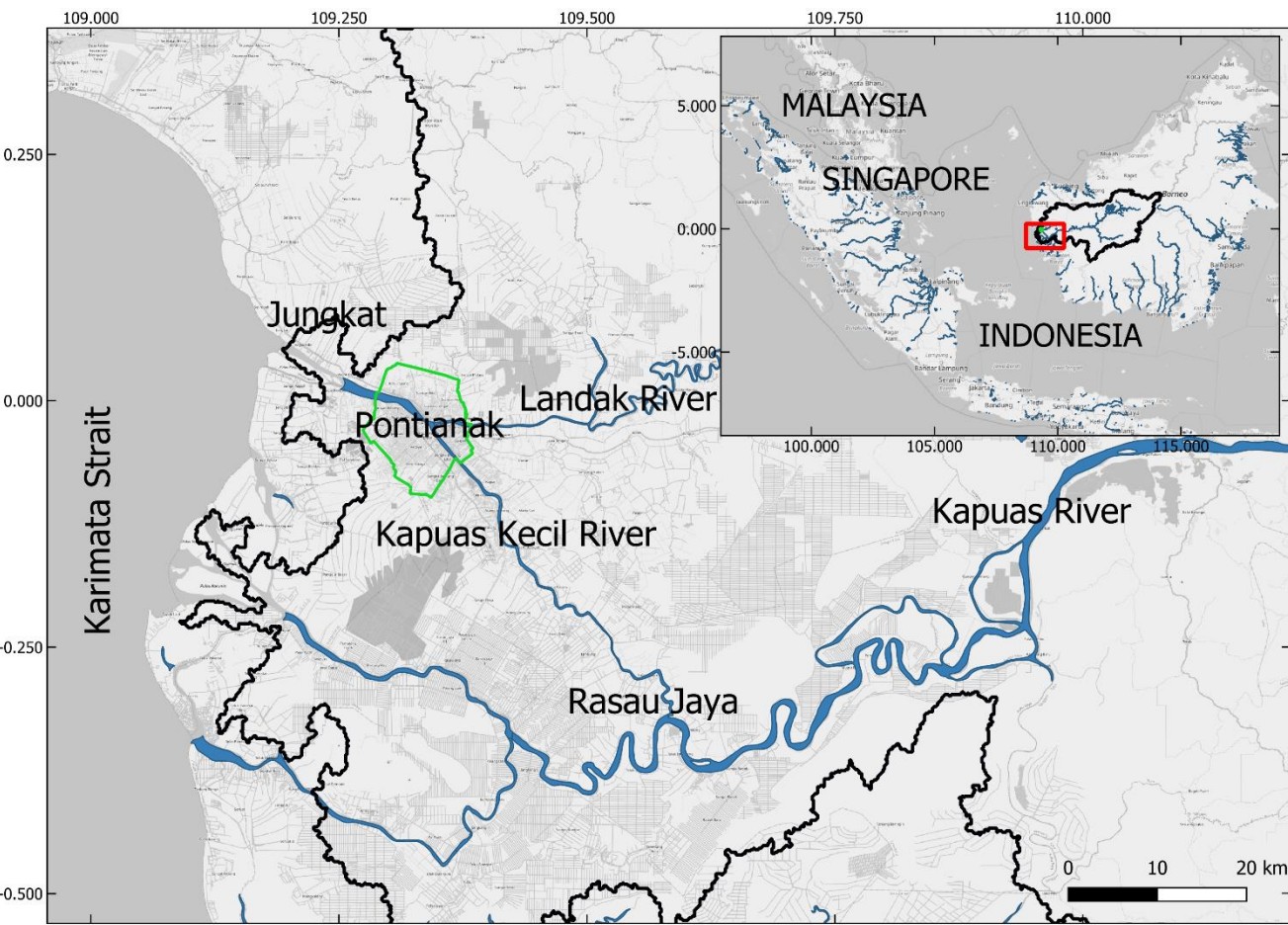


**Figure 1**: The region of interest (ROI), where the green enclosed perimeter represents the city of Pontianak. The solid black line represents the Kapuas River Watershed in the inset map, and the blue lines represent waterbodies. Background map retrieved from (Planet dump retrieved from https://planet.osm.org, 2020). © OpenStreetMap contributors 2017. Distributed under the Open Data Commons Open Database License (ODbL) v1.0.



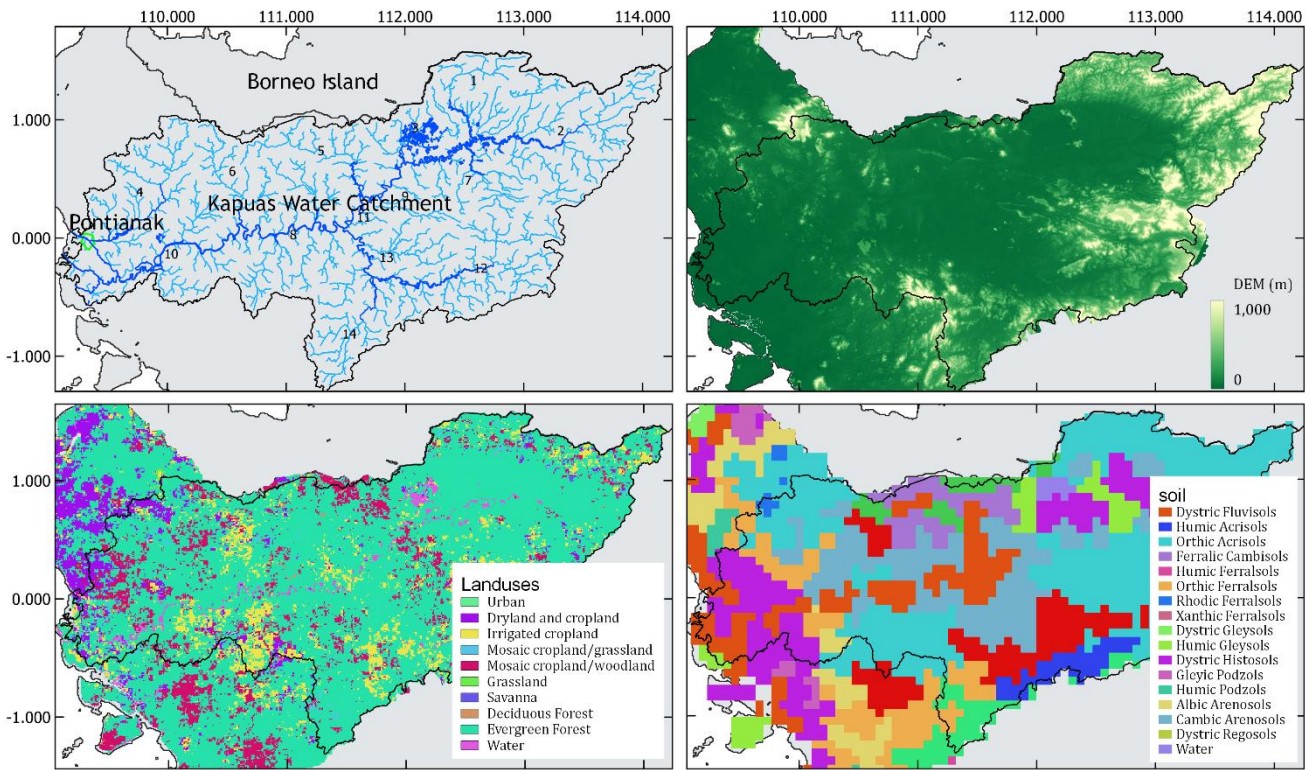

**Figure 2**: Kapuas water catchment area (upper left), Digital elevation map (upper right) retrieved from SRTM (Farr et al., 2007), Land cover maps (lower left) retrieved from CGLOPS1 (Buchhorn et al., 2020), and Soil type maps (lower right) retrieved from FAO (Sanchez et al., 2009) for the Kapuas River catchment area.

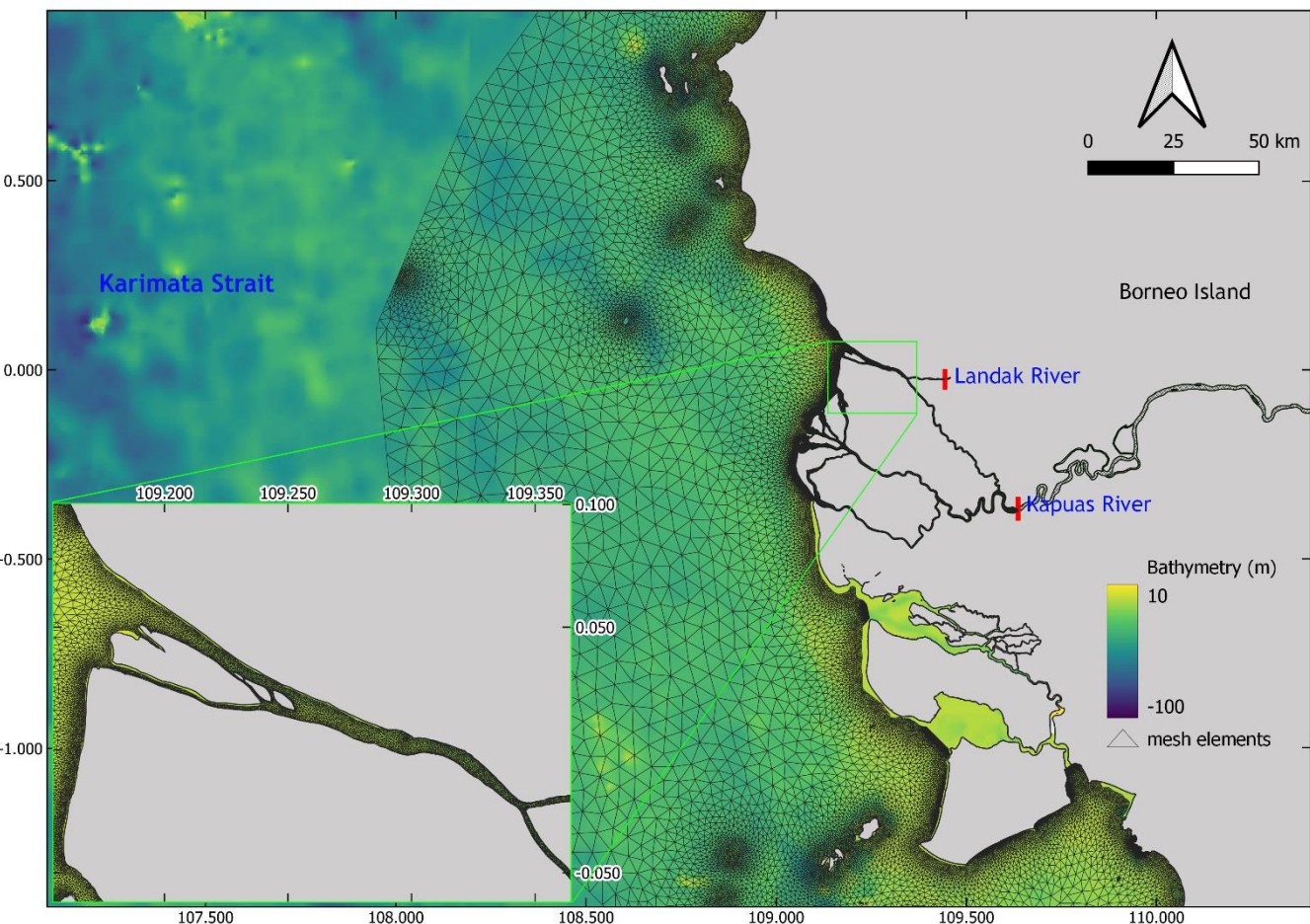

**Figure 3**: The hydrodynamic model domain is discretized with an unstructured mesh whose resolution is set to 50 m along the riverbanks, 400 m along the coast near the estuary, 1 km over the rest of the coastline, and 5 km offshore. The bathymetry of the model domain ranges from ~100 m depth offshore to 1 m in the river mouth.

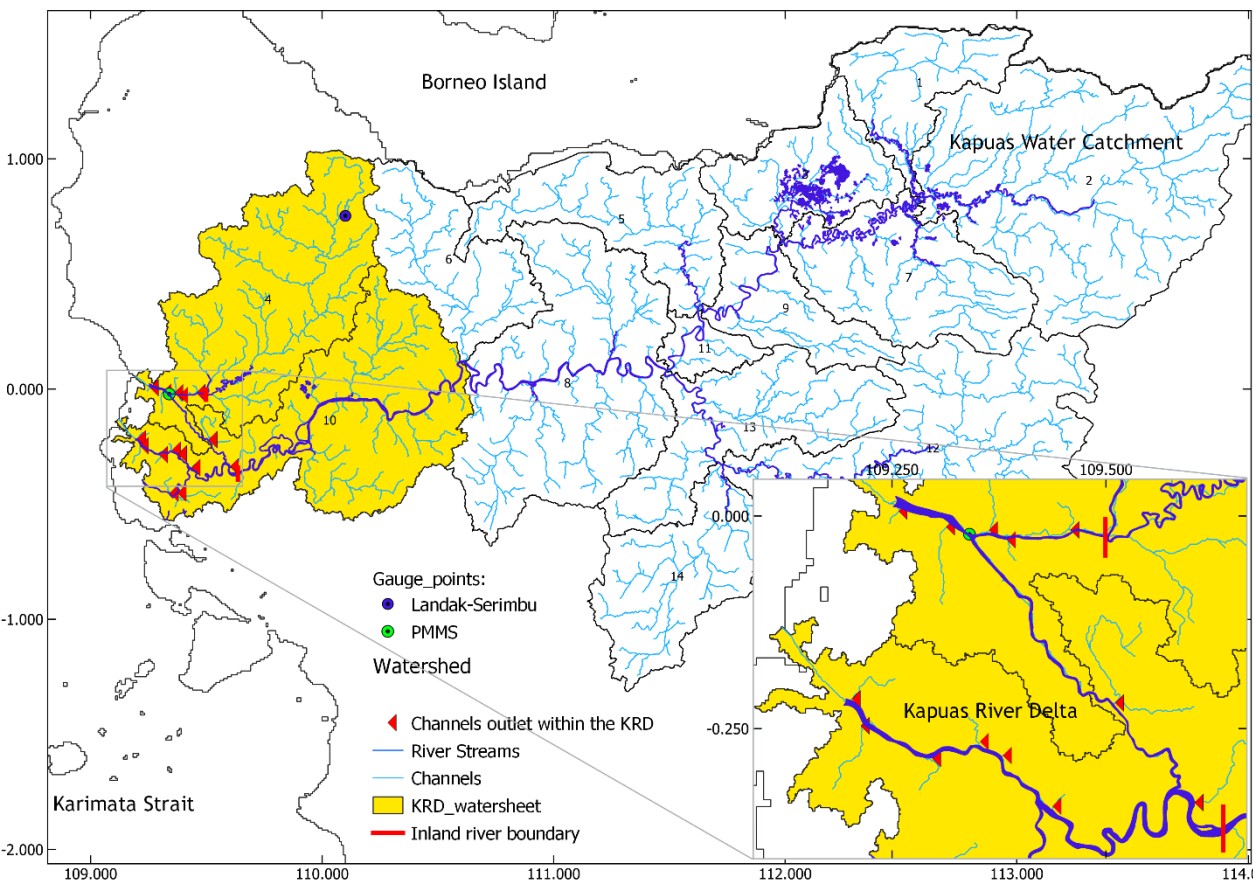

**Figure 4**: The Kapuas River watershed and its sub-basins. Since the discharges of the Kapuas River are retrieved at the middle stream, only two sub-basins are considered for the SWAT+ model (yellow area). The runoffs (channel outlets of the SWAT+ model that enter the river stream within the KRD) are set as inlets for the hydrodynamic model domain.

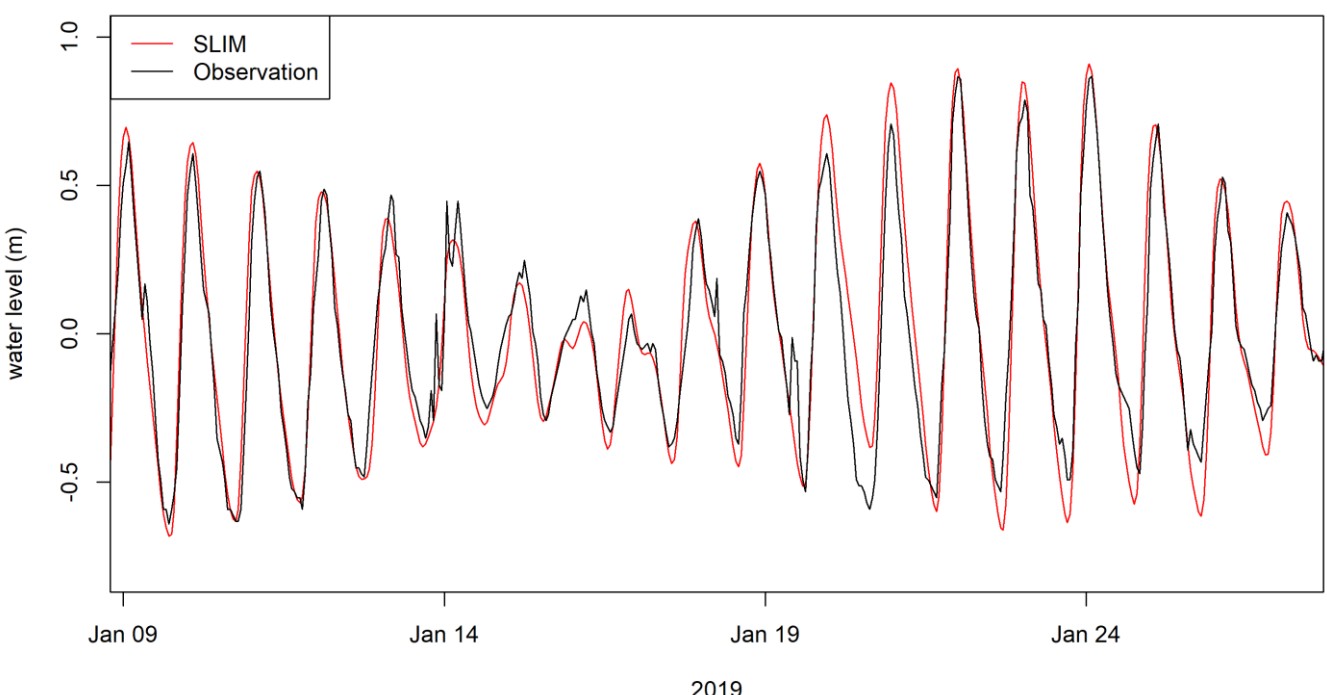

**Figure 5:** SLIM model output validation with respect to observational data at Pontianak in January 2019, with NSE = 0.87 and RMSE = 0.12 m, indicates that the model has satisfactory performance.

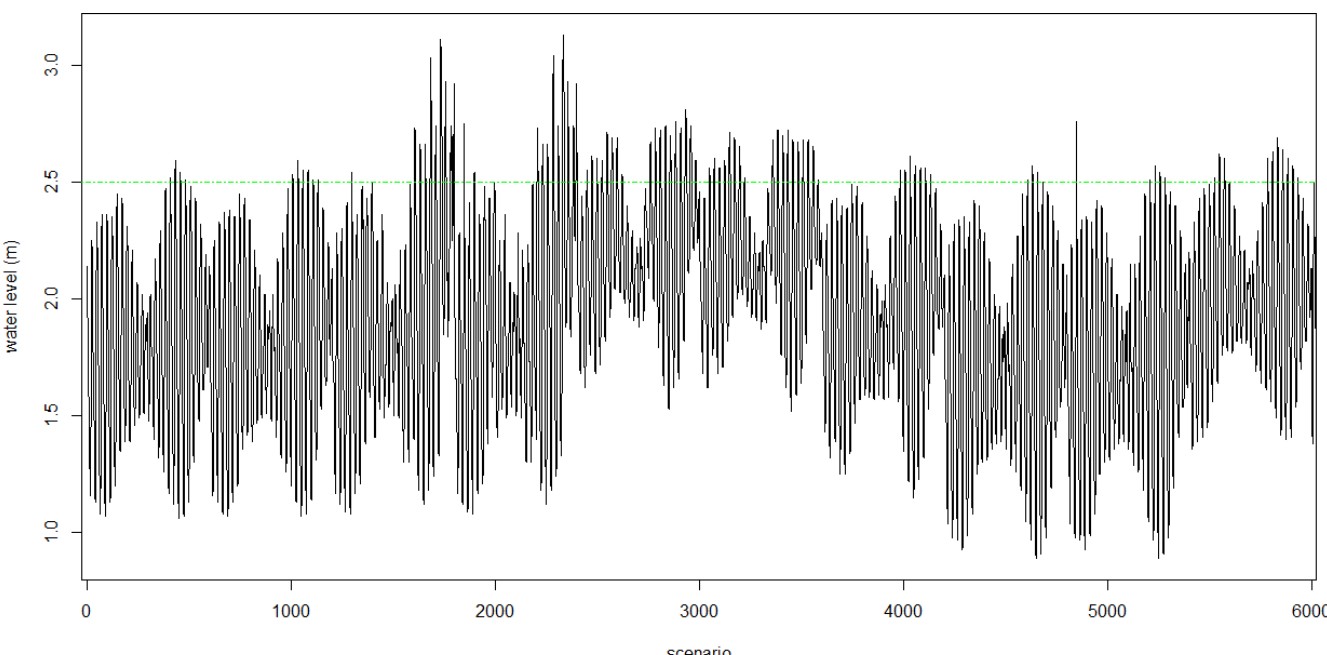

**Figure 6:** The Kapuas Kecil River's water level in Pontianak, obtained from the hydrodynamic model. The green dash line is the threshold above which the water starts to overflow the riverbanks in Pontianak.

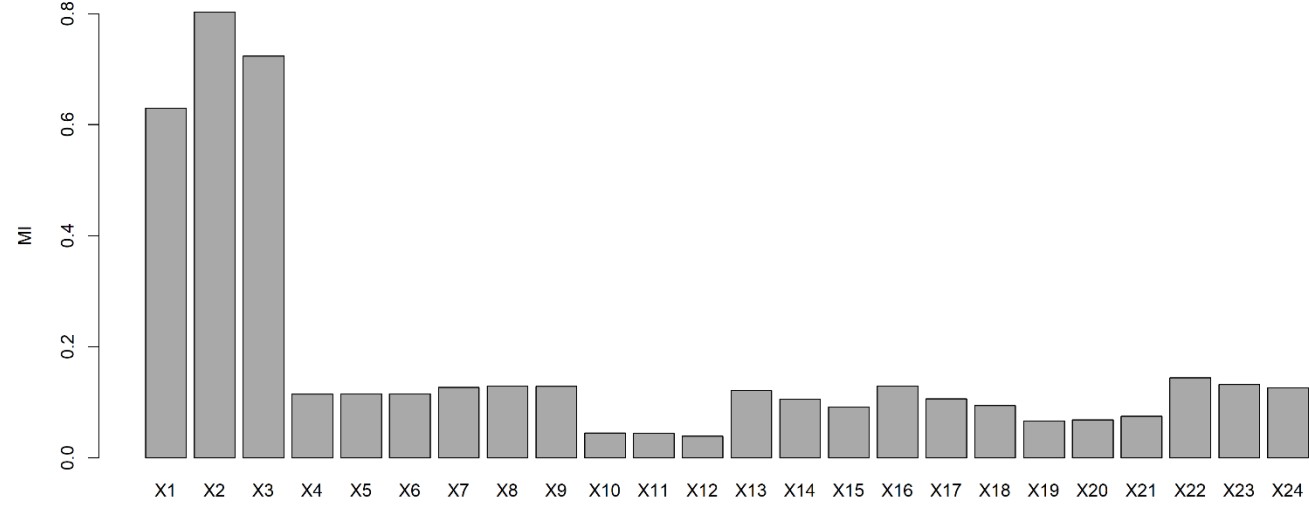


**Figure 7:** Mutual information of all predictor variables to hourly water level dynamics in 3 months of observational data.

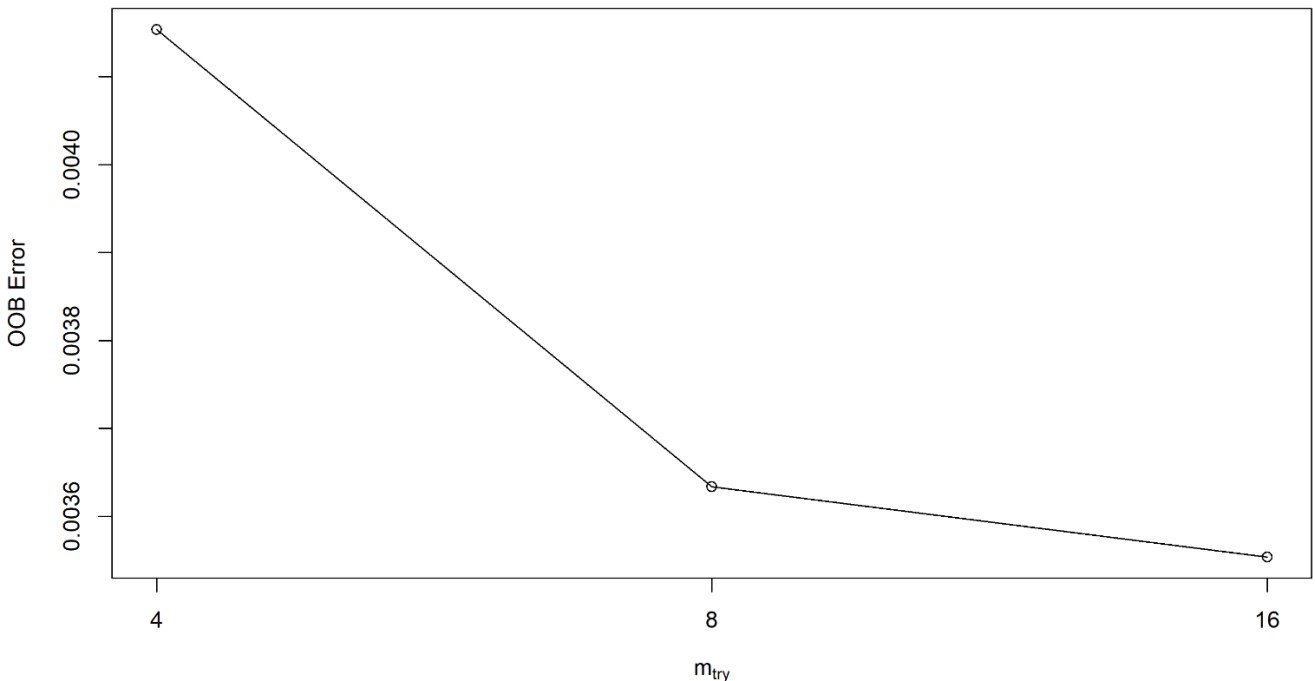

**Figure 8:** Tuned randomForest algorithm for the optimal number of variables randomly sampled as candidates at each split (mtry) parameter.

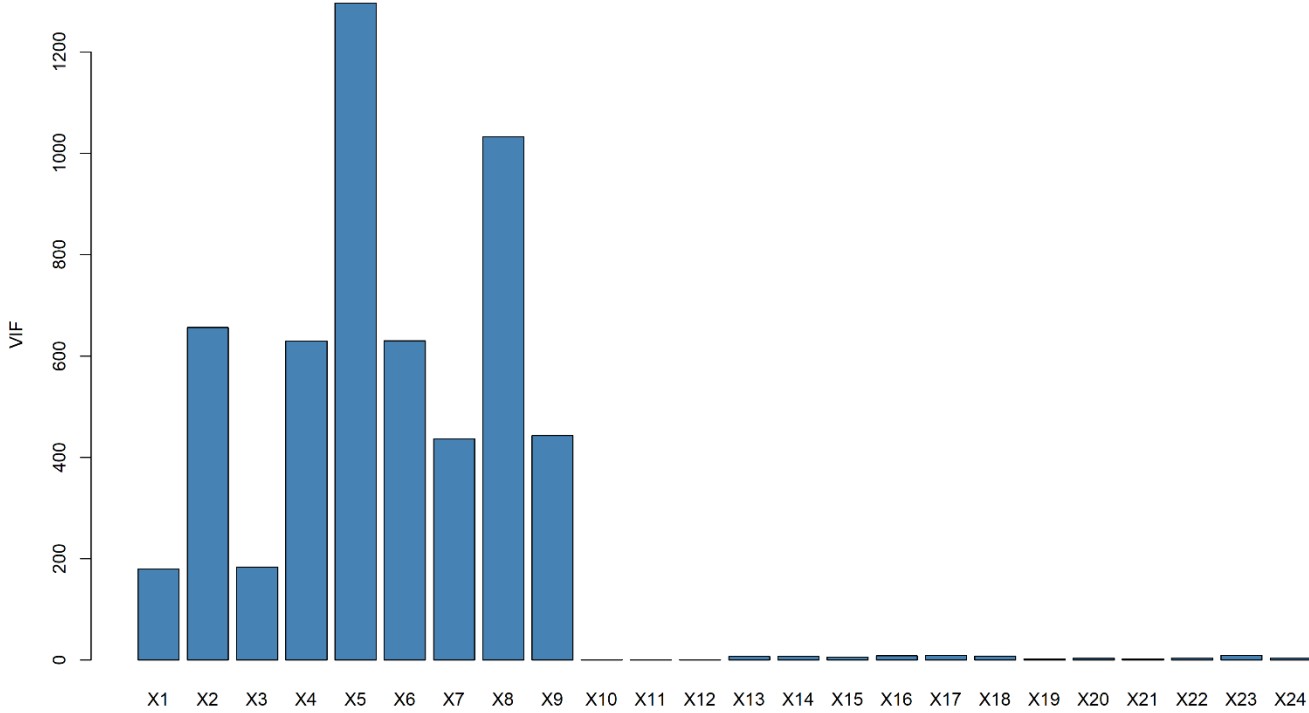

**Figure 9:** Variance Inflation Factor values of all predictor's variables in 3 months of observational data.

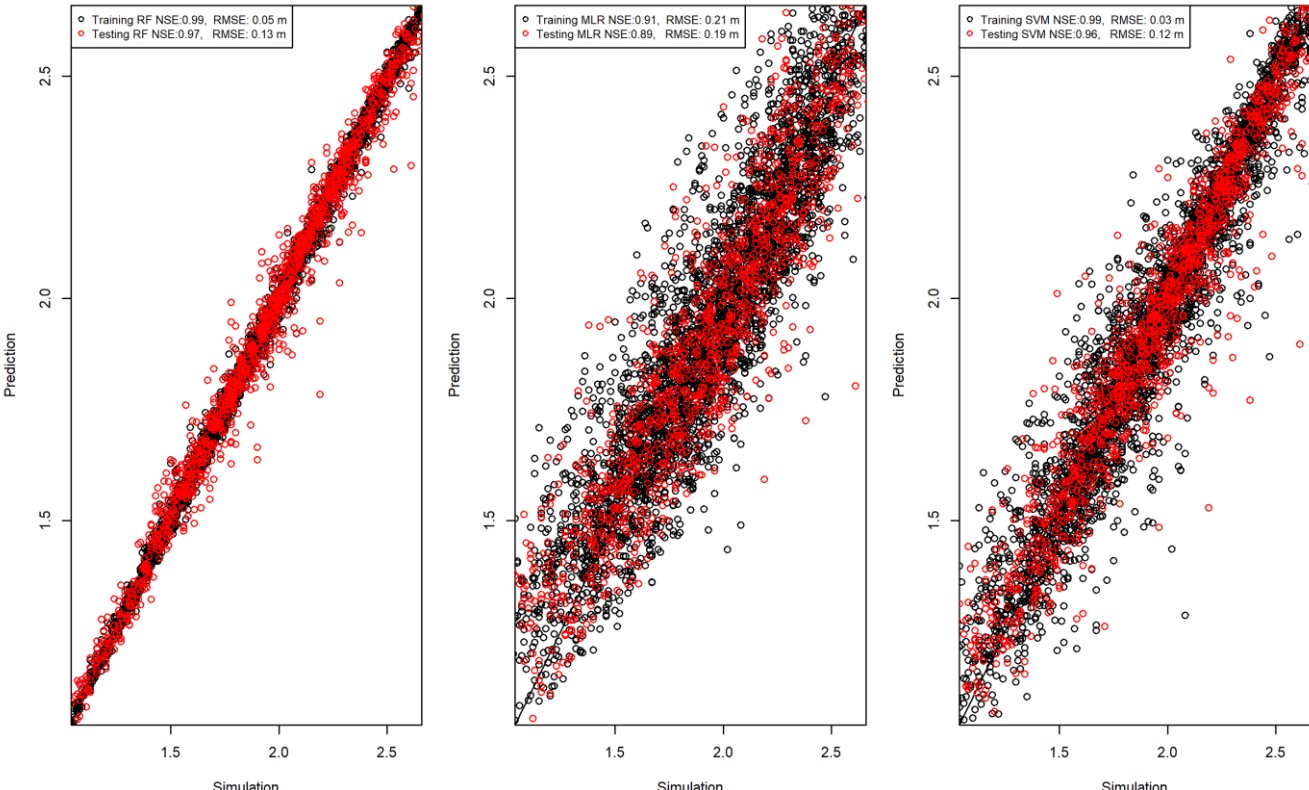

**Figure 10:** Comparison of predicted and simulated hourly water levels of the training data.

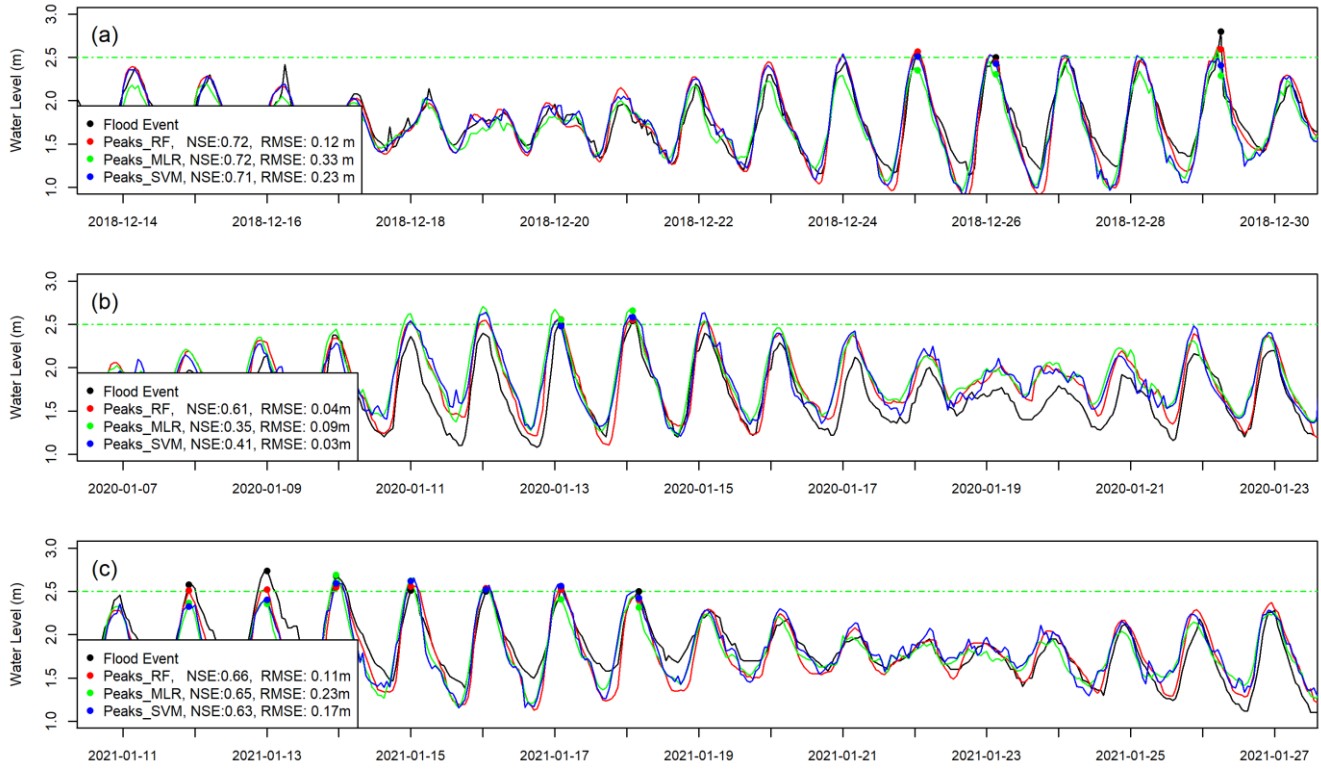

**Figure 11:** Comparison of predicted hourly water levels models and measured hourly water levels for the implementation phase on: (a)December 2018, (b) January 2020, and (c) January 2021

**Table 1.** Scenarios used to force the process-based hydrodynamic model

| n-dataset | Wind Speed (ms$^{-1}$) | Wind Direction ($^o$) | Pressure (kPa) | Discharge Kapuas (m$^3$s$^{-1}$) | Discharge Landak (m$^3$s$^{-1}$) | Rainfall (mm) |
|---|---|---|---|---|---|---|
| 1-600 | 2 - 8 | 0 – 360 | 100.5 - 101.5 | $6 \times 10^3$ | 600 | 0 |
| 601-1200 | 4-16 | 0 – 360 | 100.5 - 101.5 | $6 \times 10^3$ | 600 | 0 |
| 1201-1800 | 8-32 | 0 – 360 | 100.5 - 101.5 | $6 \times 10^3$ | 600 | 0 |
| 1801-2400 | 2 – 8 | 0 – 360 | 100.5 - 101.5 | $10^4 - 1.5 \times 10^4$ | 600 | 0 |
| 2401-3000 | 2 – 8 | 0 – 360 | 100.5 - 101.5 | $6 \times 10^3$ | 800-2100 | 0 |
| 3001-3600 | 2 – 8 | 0 – 360 | 100.5 - 101.5 | $10^4 - 1.5 \times 10^4$ | 800-2100 | 0 |
| 3601-4200 | 5 – 20 | 0 – 360 | 100.5 - 101.5 | $10^4 - 1.5 \times 10^4$ | 800-2100 | 0 |
| 4201-4800 | 2 – 8 | 0 – 360 | 100.5 - 101.5 | $3.3 \times 10^3 – 5 \times 10^3$ | 250-700 | 0 |
| 4801-5400 | 2 – 8 | 0 – 360 | 100.5 - 101.5 | $3.3 \times 10^3 – 5 \times 10^3$ | 250-700 | 0 – 150 |
| 5401-6000 | 8-32 | 0 – 360 | 100.5 - 101.5 | $6 \times 10^3$ | 600 | 0 – 150 |

**Table 2.** The variables which used as the predictors in this study.

| Code | Variable Description |
|------|---------------------|
| X1 | Tidal Elevation at Kapuas Kecil river mouth (m) |
| X2 | Tidal Elevation at Kapuas Kecil river mouth 1 hour before (m) |
| X3 | Tidal Elevation at Kapuas Kecil river mouth 2 hours before (m) |
| X4 | Hourly Discharge of the Kapuas River at Rasau Jaya in time ($m^3s^{-1}$) |
| X5 | Hourly Discharge of the Kapuas River at Rasau Jaya 1 hour before ($m^3s^{-1}$) |
| X6 | Hourly Discharge of the Kapuas River at Rasau Jaya 2 hours before ($m^3s^{-1}$) |
| X7 | Hourly Discharge of the Landak River at Kuala Mandor in time ($m^3s^{-1}$) |
| X8 | Hourly Discharge of the Landak River at Kuala Mandor 1 hour before ($m^3s^{-1}$) |
| X9 | Hourly Discharge of the Landak River at Kuala Mandor 2 hours before ($m^3s^{-1}$) |
| X10 | Hourly Precipitation at the time (mm) |
| X11 | Hourly Precipitation one hour before (mm) |
| X12 | Hourly Precipitation two hours before (mm) |
| X13 | Hourly Average Wind Speed at the time ($ms^{-1}$) |
| X14 | Hourly Average Wind Speed one hour before ($ms^{-1}$) |
| X15 | Hourly Average Wind Speed two hours before ($ms^{-1}$) |
| X16 | Hourly Maximum Instantaneous Wind Speed at the time ($ms^{-1}$) |
| X17 | Hourly Maximum Instantaneous Wind Speed one hour before ($ms^{-1}$) |
| X18 | Hourly Maximum Instantaneous Wind Speed two hours before ($ms^{-1}$) |
| X19 | Hourly Average Wind Direction at the time (degree, in the range: 0 - 360) |
| X20 | Hourly Average Wind Direction one hours before (degree, in the range: 0 - 360) |
| X21 | Hourly Average Wind Direction two hours before (degree, in the range: 0 - 360) |
| X22 | Hourly Atmospheric Pressure at the time (millibars) |
| X23 | Hourly Atmospheric Pressure one hour before (millibars) |
| X24 | Hourly Atmospheric Pressure two hours before (millibars) |

**Table 3.** Performance of the three machine learning algorithms on implementation phase

| Goodness of Fit | RF | MLR | SVM |
|-----------------|-----|-----|-----|
| NSE Training | 0.99 | 0.91 | 0.99 |
| NSE Testing | 0.97 | 0.96 | 0.89 |
| NSE Implementation1 (Dec 2018) | 0.72 | 0.72 | 0.71 |
| NSE Implementation2 (Jan 2020) | 0.61 | 0.35 | 0.41 |
| NSE Implementation3 (Jan 2021) | 0.66 | 0.65 | 0.63 |
| Total of flood predicted (out of 17 events) | 11 | 6 | 10 |
| Percent of flood predicted | 65% | 35% | 59% |
| False Positive (event) | 4 | 3 | 3 |
