# Peer review of "Integrated hydrodynamic and machine learning models for compound flooding prediction in a data-scarce estuarine delta"

_Nonlinear Processes in Geophysics, 2021_

## Referee Comment (RC1)

**Summary**

This study combines hydrodynamic modeling and machine learning techniques to predict water level in the Kecil-Kapuas River, Indonesia. The authors run 6000 hydrodynamic simulations of synthetic events and create a dataset of model outputs to train multilinear regression, random forest, and support vector machine algorithms. The trained algorithms are then used to predict water level for combinations of riverine, oceanic, and atmospheric forcing. The proposed scheme is validated against three historical flood events observed in Pontianak and suggests that random forest is the most suitable algorithm for predicting compound flooding.

**General comments**

This study develops a scheme to integrate physically-based and data-driven approaches as proposed in similar studies with either conventional machine learning algorithms (Hosseiny et al., 2020; French et al., 2017; Kabir et al., 2020b) or more advanced deep learning techniques (Kabir et al., 2020a; Muñoz et al., 2021). Yet, the scheme has critical technical flaws and is not well-aligned with the main goal of this study that consists of reducing computation burden for water level prediction in data-scarce regions. At this point, I believe that the manuscript requires major improvements to be considered for publication. Nevertheless, please find below a list of suggestions that can help improve the proposed scheme as well as my concerns that should be addressed and/or clarified.

**Specific comments**

**Title:** 'compound flooding prediction' certainly draws the reader's attention, but unfortunately there is no formal analysis in this study suggesting that Pontianak experienced such compound events in the past. Although the introduction elaborates on the mechanisms triggering compound flooding, the authors should conduct preliminary process-based or statistical analyses to confirm this. See for example the works of Kumbier et al., (2018), Valle-Levinson (2020), Ward et al., (2018), Ghanbari et al., (2021) among others.

**Abstract:** 'Compound flood' scenarios should be derived from preliminary statistical analysis that account for the dependence among flood drivers. See for example Serafin et al., (2019), Moftakhari et al., (2019), etc. The abstract should be revised to clearly report the findings of this study. Random forest is used to predict water levels only, yet 'flooding hazards' and 'compound flooding' are mentioned repeatedly without presenting any flood maps in Pontianak. A very preliminary assessment could be achieved by projecting water level to adjacent areas of the Kapuas-Kecil River in case 2D flood modeling is computationally demanding.

**Introduction:** Literature review falls short in content and cohesion. The introduction should mention the state-of-the-art techniques for compound flood hazard modeling and assessment. (Bevacqua et al., 2019; Couasnon et al., 2020; Ye et al., 2021; Muñoz et al., 2022).

**Material and methods:** This section is of major concern since the methodology presented in this study is not technically sound. I agree that developed countries might face challenges to implement hydrodynamic models due to data scarcity and computational resource limitations. Nevertheless, assuming that those countries develop such a model and want to implement the proposed scheme.

Are 6000 simulations really necessary to train machine learning algorithms? Would not it be better to wisely sample a small set of realistic forcing conditions that effectively lead to compound flooding? Although machine learning is a 'data-hungry' technique, I consider thousand of hydrodynamic simulations a bit exaggerated. I suggest the authors train the models with a small sample size (e.g., hundred of simulations) and report the results. This can help reduce computation time associated with hydrodynamic simulations. How long does it take to run the 2D-model of the Kapuas-Kecil in a regular desktop computer?

**Section 2.2:** There is no information regarding model calibration. This is critical as the authors rely on hydrodynamic simulations to train the machine learning algorithms. Referring to a pre-print/unpublished work (Sampurno et al., 2021) for additional details of the model is not acceptable. Please describe the model calibration process in detail.

**Section 2.4.** Another point of major concern is the calibration of machine learning algorithms. It is no clear whether the authors tuned random forest and support vector machine in the training phase or not. In that regard, the training dataset (e.g., 6000 model outputs with the associated input variables) should have been split into training/validation datasets to conduct hyperparameter tuning and so prevent overfitting issues. Using all model outputs to train the algorithms (as reported here) and relying on default parameter-values is not a wise use of machine learning (e.g., Random requires tuning of the number of trees, sample leaf, sample split, etc.). The authors should conduct a thorough 'hyperparameter' tuning as it substantially improves the performance of machine learning algorithms.

**Results.** "Even though all algorithms perform very well during the training phase, the performances are different during the testing phases". This is known as overfitting (Ying, 2019) and occurs because random forest and support vector machine are not calibrated/tuned in the training phase.

**Technical corrections**

L10 and thorough the text: There are odd terms that should be corrected like hydrodynamic modeling instead of 'water level modeling'.

L28: What is the growth rate in the last decade?

L38: Please elaborate more on non-structural measures. This sentence is not clear.

L40: Which issue? Please explain clearly.

L45-47: 'Machine learning can enable us…' How? Please, elaborate more on this. More references are needed discussing the benefits of machine learning for water level prediction and/or flood forecasts.

L96: Please locate Pontianak in Figure 2.

L97: Section 2.3 is very short (4 lines) and should be included in the previous section.

L65: More details of the study area are needed. What is the catchment size, average river flow, tidal regime, rate of local sea level rise at the Kapuas River?

L115: I suggest a more robust statistical analysis to evaluate multicollinearity among the variables (e.g., Variance Inflation Ratio (VIF), see Alipour (2020)). Multicollinearity negatively affects the performance of support vector machine and multilinear regression models. VIF can help reduce the number of predictors.

L155-156: NSE and RMSE might improve after a thorough hyperparameter tuning of the machine learning algorithms.

L158: There are no such inundation scenarios (no flood maps). This should be clarified and better replaced for water level scenarios.

Table 1. What are the criteria to come up with those range of values?

Figure 2. Scale bar and north arrow are missing.

Figure 6. Comparison of predicted and 'simulated' hourly water levels of training data. There are no observed water levels in the training phase.

Figure 8. X-axis is not observation but hydrodynamic simulation.

L295. There are references not included in the main text. See for example Rozum et al., 2020.

**References**

Alipour, A., Ahmadalipour, A., Abbaszadeh, P., and Moradkhani, H.: Leveraging machine learning for predicting flash flood damage in the Southeast US, Environ. Res. Lett., 15, 024011, https://doi.org/10.1088/1748-9326/ab6edd, 2020.

Bevacqua, Maraun, D., Vousdoukas, M. I., Voukouvalas, E., Vrac, M., Mentaschi, L., and Widmann, M.: Higher probability of compound flooding from precipitation and storm surge in Europe under anthropogenic climate change, 5, eaaw5531, https://doi.org/10.1126/sciadv.aaw5531, 2019.

Couasnon, A., Eilander, D., Muis, S., Veldkamp, T. I. E., Haigh, I. D., Wahl, T., Winsemius, H. C., and Ward, P. J.: Measuring compound flood potential from river discharge and storm surge extremes at the global scale, Nat. Hazards Earth Syst. Sci., 20, 489–504, https://doi.org/10.5194/nhess-20-489-2020, 2020.

French, J., Mawdsley, R., Fujiyama, T., and Achuthan, K.: Combining machine learning with computational hydrodynamics for prediction of tidal surge inundation at estuarine ports, Procedia IUTAM, 25, 28–35, https://doi.org/10.1016/j.piutam.2017.09.005, 2017.

Ghanbari, M., Arabi, M., Kao, S.-C., Obeysekera, J., and Sweet, W.: Climate Change and Changes in Compound Coastal-Riverine Flooding Hazard Along the U.S. Coasts, 9, e2021EF002055, https://doi.org/10.1029/2021EF002055, 2021.

Hosseiny, H., Nazari, F., Smith, V., and Nataraj, C.: A Framework for Modeling Flood Depth Using a Hybrid of Hydraulics and Machine Learning, 10, 8222, https://doi.org/10.1038/s41598-020-65232-5, 2020.

Kabir, S., Patidar, S., Xia, X., Liang, Q., Neal, J., and Pender, G.: A deep convolutional neural network model for rapid prediction of fluvial flood inundation, Journal of Hydrology, 590, 125481, https://doi.org/10.1016/j.jhydrol.2020.125481, 2020a.

Kabir, S., Patidar, S., and Pender, G.: A machine learning approach for forecasting and visualising flood inundation information, 1–15, https://doi.org/10.1680/jwama.20.00002, 2020b.

Kumbier, K., Carvalho, R. C., Vafeidis, A. T., and Woodroffe, C. D.: Investigating compound flooding in an estuary using hydrodynamic modelling: a case study from the Shoalhaven River, Australia, 18, 463–477, https://doi.org/10.5194/nhess-18-463-2018, 2018.

Moftakhari, Schubert, J. E., AghaKouchak, A., Matthew, R. A., and Sanders, B. F.: Linking statistical and hydrodynamic modeling for compound flood hazard assessment in tidal channels and estuaries, Advances in Water Resources, 128, 28–38, https://doi.org/10.1016/j.advwatres.2019.04.009, 2019.

Muñoz, D. F., Muñoz, P., Moftakhari, H., and Moradkhani, H.: From local to regional compound flood mapping with deep learning and data fusion techniques, Science of The Total Environment, 782, 146927, https://doi.org/10.1016/j.scitotenv.2021.146927, 2021.

Muñoz, D. F., Abbaszadeh, P., Moftakhari, H., and Moradkhani, H.: Accounting for uncertainties in compound flood hazard assessment: The value of data assimilation, Coastal Engineering, 171, 104057, https://doi.org/10.1016/j.coastaleng.2021.104057, 2022.

Serafin, K. A., Ruggiero, P., Parker, K., and Hill, D. F.: What's streamflow got to do with it? A probabilistic simulation of the competing oceanographic and fluvial processes driving extreme along-river water levels, 19, 1415–1431, https://doi.org/10.5194/nhess-19-1415-2019, 2019.

Valle-Levinson, A., Olabarrieta, M., and Heilman, L.: Compound flooding in Houston-Galveston Bay during Hurricane Harvey, Science of The Total Environment, 747, 141272, https://doi.org/10.1016/j.scitotenv.2020.141272, 2020.

Ward, P. J., Couasnon, A., Eilander, D., Haigh, I. D., Hendry, A., Muis, S., Veldkamp, T. I. E., Winsemius, H. C., and Wahl, T.: Dependence between high sea-level and high river discharge increases flood hazard in global deltas and estuaries, Environ. Res. Lett., 13, 084012, https://doi.org/10.1088/1748-9326/aad400, 2018.

Ye, F., Huang, W., Zhang, Y. J., Moghimi, S., Myers, E., Pe'eri, S., and Yu, H.-C.: A cross-scale study for compound flooding processes during Hurricane Florence, 21, 1703–1719, https://doi.org/10.5194/nhess-21-1703-2021, 2021.

Ying, X.: An Overview of Overfitting and its Solutions, J. Phys.: Conf. Ser., 1168, 022022, https://doi.org/10.1088/1742-6596/1168/2/022022, 2019.

---

## Referee Comment (RC2)

**Summary**

The article "Integrated hydrodynamic and machine larning models for compound flooding prediction in a data-scarce estuarine delta" describes a framework based on deterministic hydrodynamic modeling and artificial intelligence (A.I.) to estimate compound flood events. The authors assessed three different techniques of machine learning (i.e., A.I. model) using the results from the hydrodynamic model, which used thousands (e.g., 6,000) different environmental forcings combinations. The compound flood event described in this manuscript is based on coastal (e.g., wind velocity, atmospheric pressure, and astronomical tides) and fluvial and pluvial stressors. This study aims to use the results from the trained A.I. model to predict compound flood in a scarce-data region such as Pontianak, Indonesia.

**General Comments**

I will like to congratulate the authors for writing a great manuscript. This manuscript was very well put together that tells a story in an organized and scientific manner. However, some essential details were omitted from this first draft. First, the hydrodynamic model calibration/validation is missing from section 2.2. I understand that the authors reference the reader to another article (currently under review) for more details on the hydrodynamic model. But in any flood modeling, it is crucial to discuss the hydrodynamic model calibration/validation, especially when the outputs of this model will be used as input in another model. At a minimum, the authors should dedicate a paragraph (if not a subsection) to discuss the results and method of the hydrodynamic model calibration/validation without going into much detail since the authors can reference another article.

Second, the flooding scenarios selected (e.g., Table 1) used to describe compound floods using the hydrodynamic model lacks information. For example, the authors should explain why they selected that certain combination of environmental factors and related to any observations or datasets. Finally, the authors do not give any information on the coupling occurring between the different hydrodynamic models to assess compound floods. This aspect is crucial in this type of research, and at least a subsection should be dedicated to explaining these model pass information between them to account for compound floods. Nevertheless, before I can accept the article for publishing, it needs to go through a major revision that will require a re-revision from the reviewers. The authors should recall that the main purpose of publishing a research article like this is to adopt the proposed methods and apply them to their region of interest. Therefore, it is crucial to include as much detail as pertinent to replicate the proposed work. Please find below some specific comments and questions that need to be addressed in the revised version of the author's manuscript.

**Specific Comments**

**Section 1**

- Line 25: the authors should include the following publications as part of this citation: Santiago-Collazo et al. (2021), Gori et al. (2022), Hsiao et al. (2021), Ghanbari et al. (2021)
- Line 26: the authors should include the following publications as part of this citation: Ikeuchi et al. (2017), Wahl et al. (2015)

- It needs a paragraph of a literature review of previous modeling frameworks that uses a deterministic model to train an A.I. model. This will help put in context to the reader earlier attempts of this modeling approach. This might be the first attempt to simulate compound flood events, but other studies might focus on different processes such as subsurface flow and even at other disciplines such as transportation and structural engineering. Some questions that can be answered from including this paragraph might be the following:
  - Is this the first study that uses a deterministic-A.I. modeling framework to estimate compound floods? If not, how was it then, and what was their approach?
  - Have other researchers used a deterministic-A.I. modeling framework to estimate different parameters outside of surface flow physics?

**Section 2.1**

- It will be beneficial for the reader to include an additional figure with the study area's topographic/bathymetric elevation map and land use/land cover maps and soil type maps since all these parameters will affect surface runoff modeling than subsequently will affect the compound flood magnitude. If there is no such data available as a map format, the authors should indicate it in the manuscript. This will highlight the date scarcity in the region.
- Consider adding the Kapuas River watershed area and compare it with the total island extent. This will help the reader put the extension of this watershed into context, rather than just saying that it is the longest island river.
- Figure 1: need to include in the figure caption that the solid black line represents the Kapuas River Watershed on the insert map. Also, mention that the blue lines represent waterbodies.

**Section 2.2**

- Line 76: need to add a reference to cite the SLIM 2d hydrodynamic model. Similar to the SWAT+ citation on Line 94
- Need to add a paragraph or subsection of the calibration/validation of both deterministic models used: SLIM 2D and SWAT+
- Line 85-89: How far inland does the mesh extend through the river? Does it penetrate through the riverine floodplain or stop at the river bank's height? Does the digital elevation model (DEM) used in the hydrodynamic model (details are not given) penetrate beneath the water to capture the full river bathymetry (i.e., description of the terrain surface underwater), so the riverine cross-section is described fully, or does it reflect the water surface elevation? If the complete riverine cross-section is not available from observed data, which cross-sectional area do the authors use? These details are not given in the text nor Figure 2.
- Figure 2: include a bathymetry elevation as a color-filled contour with the unstructured mesh, so the reader can examine if there are any canyons or through underwater that will affect the coastal processes flood modeling. The authors may also consider adding the mesh resolution like a color map, see Figure 3 on Bislkie et al. (2020).
- Line 90-92: information about the different environmental factors considered in the study was given in Table 1. However, information regarding the astronomical tide forcing is not given, just from the model that was obtained. I think that more information should be given since, at the discussion session, the authors concluded that tidal forcing is the factor that

most affects the compound flood levels in the regions. The authors should answer the following questions within the text:

- o What is the average tidal amplitude (e.g., micro-tidal, meso-tidal, macro-tidal)?
- o What is the dominant tidal constituent (e.g., M2, S1, K1, etc.)?
- o What is the tidal regime (i.e., period) at the region (e.g., diurnal, semi-diurnal, or mixed )?

- Line 92-93: the authors should explain in more detail the coupling procedure between SWAT+ and SLIM 2D. Also, the authors should locate on a map the riverine boundary conditions in the SLIM 2D model and clearly specify the total amount of locations. The following questions should be answered in the text of the manuscript:
  - o What type of coupling is occurring between the models (e.g., one-way, two-way, tightly, or fully coupling)?
  - o How often (e.g., each computational time step) does the exchange of information happen?
  - o Do the SWAT+ model runs first and independently, and once it finishes the simulation, it passes the information to SLIM 2D, or do both models run simultaneously?
  - o Is the location of the riverine boundary conditions in the SLIM 2D model inland enough (i.e., away from the coast) that coastal processes will not affect the water levels? If not, the authors should justify the selection of that location.

- Line 93-95: the authors do not give any information regarding the hydrologic modeling using SWAT+. Since it does not reference another publication, at least a subsection should be dedicated to providing more details of this model. This information is crucial since the SWAT+ model computes the pluvial and fluvial processes in the compound flood simulation in SLIM 2D. For example, Silva-Araya et al. (2018) described their hydrologic and hydrodynamic model in separate subsections before describing the coupling technique in an additional subsection. The following questions should be answered in the text of the manuscript:
  - o Does infiltration processes are taken into consideration?
  - o How many sub-watershed was the Kapuas River watershed divided into so it was suitable to model in SWAT+?
  - o What is the extent of the SWAT+ model? A figure should be included.
  - o What was the temporal resolution of this model?
  - o Did the rainfall vary in time and space through the domain?

- Line 95-96: the authors should include in Figure 1 (or on an additional figure) the location of the gauge where the observational data was obtained. What type of observational data was used to evaluate the model performance (e.g., stage, discharge, high-water marks, etc.)?

**Section 2.3**

- This section lacks much essential information for the reader, and it is not clear. This section is one of the most important in the manuscript since it will control the compound flood event being simulated. The following questions should be answered in the text of the manuscript:
  - o Table 1:
    - ▪ From where were these values chosen, and why these values themselves?

- ▪ Why do the tables display only a single value of discharge, whereas, in Line 109, the authors said that the datasets (including riverine variables) were recorded hourly? Is the value shown in the table represents the annual peak discharge, the average value, etc.?
  - o Why 6,000 simulations and not 1,000 or 10,000? Need to justify the author's decision.
  - o How was the combination of the different parameters chosen? Did the authors use any statistical approaches, such as a Monte Carlo Simulation, or used a random distribution?
  - o Why the hydrodynamic model was run for 10 months and not 12 or 6 ?

**Section 2.4.1**

- Line 108-109: why did the authors select just one and two hours before the flood event as the prior conditions? It has been shown that rainfall events that occur three days before a flood event have measurable effects on the compound flood levels (Bilskie et al., 2021). The authors need to justify their selection. What was the SLIM output temporal resolution?
- Table 2: the biggest tidal variations occur within 6 to 12 hours before/after their peak level, depending on the tidal regime. Therefore, it does not make sense to vary their tidal elevation (which is not given in Table 1 nor the text) by one or two hours since the values are very similar. It will make more sense that the authors tested scenarios that considered the high and low tidal elevation, which can be 6 to 12 hours apart.

**Section 3**

- Line 155-157: need to cite other studies that confirm your statement that a model with those values of NSE and RMSE is a "good proxy" of the real system.
- Line 158-161: this can be moved to Section 2.
- Figure 5: the authors should comment if the low impact of rainfall to compound flood events might be related to the small amount of rainfall used. Also, can the selection of a lumped-parameter hydrologic model (SWAT+) used in this study affect the surface runoff quantity fed into the hydrodynamic model?
- Add a vertical axis label to Figure 5.
- Improve the resolution of Figures 3, 6, and 7.

**Section 4**

- The authors should comment if the low accuracy of the A.I. model during the testing phase is related to the calibration/validation of the hydrodynamic model? If the hydrodynamic model is inaccurate in predicting real-life floods, then the A.I. model will have low accuracy.
- Why is the biggest impact of the compound flood levels due to tidal conditions? How do these findings relate to the physical processes occurring at this location? Have other studies drawn similar conclusions regarding the importance of tides in a compound flood event? The authors should talk more about this.

**Section 5**

- The conclusion session needs improvement. For example, topics in the discussion section should be at the conclusion section, such as modeling limitations and future research.

- The authors should also include as part of their modeling limitation the use of the SWAT+ model to quantify the pluvial and fluvial processes in their compound flood event. The SWAT+ model is a conceptual-based, lumped-parameter hydrologic model. Therefore, this model has many limitations when computation spatially- and time-varying surface flow compared to physically-based, distributed-parameter hydrologic models capable of having a spatial distribution of precipitation and watershed properties through a computational grid.

**Technical Corrections**

- Line 111: it should say "Statistic tool" and not "atistic tool"

**References**

A. Gori et al. (2022). "Tropical cyclone climatology change greatly exacerbates U.S. extreme rainfall–surge hazard." Nature Climate Change.

S. Hsiao et al. (2021). "Flood risk influenced by the compound effect of storm surge and rainfall under climate change for low-lying coastal areas." Science of the Total Environment.

F. L. Santiago-Collazo et al. (2021) "An Examination of Compound Flood Hazard Zones for Past, Present and Future Low-gradient Coastal Land-margins." Frontiers in Climate Change.

M. Ghanbari et al. (2021). "Climate Change and Changes in Compound CoastalRiverine Flooding Hazard Along the U.S. Coasts." Earth's Future

T. Wahl et al. (2015). "Increasing risk of compound flooding from storm surge and rainfall for major US cities." Nature Climate Change.

H. Ikeuchi et al. (2017). "Compound simulation of fluvial floods and storm surges in a global coupled river-coast flood model: model development and its application to 2007 Cyclone Sidr in Bangladesh." Journal of Advances in Modeling Earth Systems.

M. V. Bilskie et al. (2020). "Unstructured finite element mesh decimation for real-time Hurricane storm surge forecasting." Coastal Engineering.

W.F. Silva-Araya et al. (2018). "Dynamic modeling of surface runoff and storm surge during hurricane and ropical storm events." Hydrology.

M.V. Bilskie et al. (2021). "Enhancing flood hazard assessments in coastal Louisiana through coupled hydrologic and surge processes."Frontier in Water.

---

## Author Comment (AC1)

**Response to the first reviewers' comments (RC1) on the paper** "*Integrated hydrodynamic and machine larning models for compound flooding prediction in a data-scarce estuarine delta*".

**General comments**

*This study develops a scheme to integrate physically-based and data-driven approaches as proposed in similar studies with either conventional machine learning algorithms (Hosseiny et al., 2020; French et al., 2017; Kabir et al., 2020b) or more advanced deep learning techniques (Kabir et al., 2020a; Muñoz et al., 2021). Yet, the scheme has critical technical flaws and is not well-aligned with the main goal of this study that consists of reducing computation burden for water level prediction in data-scarce regions. At this point, I believe that the manuscript requires major improvements to be considered for publication. Nevertheless, please find below a list of suggestions that can help improve the proposed scheme as well as my concerns that should be addressed and/or clarified.*

*Response:*

We want to thank the reviewer for taking the time to review our paper. Their comments are beneficial and helped us to improve the article. In what follows, we addressed the comments, where the reviewer's comments are presented in italic type and my response in roman type.

**Specific comments**

*Title: 'compound flooding prediction' certainly draws the reader's attention, but unfortunately there is no formal analysis in this study suggesting that Pontianak experienced such compound events in the past. Although the introduction elaborates on the mechanisms triggering compound flooding, the authors should conduct preliminary process-based or statistical analyses to confirm this. See for example the works of Kumbier et al., (2018), Valle-Levinson (2020), Ward et al., (2018), Ghanbari et al., (2021) among others.*

*Response:*

We did the assessment of compound flooding over the area using a 2D hydrodynamic model in the previous work (Sampurno et al., 2021). The paper related to the work was accepted for publication in the journal Biogeosciences and is expected to be available online in due course. So, this work is to continue what we have done in the previous work. As a complement, we added these sentences in the introduction session:

This city experienced a compound flooding event on 29 December 2018 (Sampurno et al., 2021), and the impact was severe (Madrosid, 2018). At that moment, the water level dynamic is about to go down after passing its peak elevation, when suddenly a strong force pushes it to go up again for a short moment. The interaction between tides, storm surges, and discharges along the tidal river in the Kapuas River delta is responsible for a 30 cm increase in the water level during the event.

***Abstract:*** *'Compound flood' scenarios should be derived from preliminary statistical analysis that account for the dependence among flood drivers. See for example Serafin et al., (2019), Moftakhari et al., (2019), etc. The abstract should be revised to clearly report the findings of this study. Random forest is used to predict water levels only, yet 'flooding hazards' and 'compound flooding' are mentioned repeatedly without presenting any flood maps in Pontianak. A very preliminary assessment could be achieved by projecting water level to adjacent areas of the Kapuas-Kecil River in case 2D flood modeling is computationally demanding.*

***Response:***

We followed the suggestion and modified the abstract. We only focus on flood prediction but not yet on a flood hazard assessment. Regarding the "compound flooding" term, the study area has the possibility of experiencing three possible flooding types (coastal, urban/flash, and riverine). Since these events generally coincide or occur in the near time in the area, therefore, compound flooding term is chosen to represent the issues.

Here is the updated abstract:

"Flood forecasting based on hydrodynamic modeling is an essential non-structural measure against compound flooding over the globe. With the risk increasing under climate change, all coastal areas are now in need of flood risk management strategies. Unfortunately, for local water management agencies in developing countries, building such a model is challenging due to the limited computational resources and the scarcity of observational data. We attempt to solve this issue by proposing an integrated hydrodynamic and machine learning approach to predict water level dynamics as a proxy of compound flooding risk in a data-scarce delta. As a case study, this integrated approach is implemented in Pontianak, the densest coastal urban area over the Kapuas River delta, Indonesia. Firstly, we built a hydrodynamic model to simulate several compound flooding scenarios. The outputs are then used to train the machine learning (ML) model. To obtain a robust machine learning model, we consider three machine learning algorithms, i.e., Random Forest, Multi Linear Regression, and Support Vector Machine. Our results show that the integrated scheme works well. The Random Forest (RF) is the most accurate algorithm to model water level dynamics in the study area. Meanwhile, the machine-learning model with the RF algorithm can predict eleven out of seventeen compound flooding events during the implementation phase. It could be concluded that RF is the most appropriate algorithm to build a reliable ML model capable of estimating the river water level dynamics within Pontianak, whose output can be used as a proxy for predicting compound flooding events in the city."

***Introduction:*** *Literature review falls short in content and cohesion. The introduction should mention the state-of-the-art techniques for compound flood hazard modeling and assessment. (Bevacqua et al., 2019; Couasnon et al., 2020; Ye et al., 2021; Muñoz et al., 2022).*

***Response:***
The suggestion has been followed. We mention these references in the introduction. Then, we also add a new paragraph as follows:

A new paradigm that combines deterministic and machine learning components has been proposed and implemented to tackle data and computational limitations in environmental modeling (Krasnopolsky and Fox-Rabinovitz, 2006; Goldstein and Coco, 2015). However, to the best of our knowledge, no previous modeling frameworks have developed a deterministic model to train a machine learning model for compound flooding studies. As a common practice, compound flood modeling typically uses the coupling of two or more hydrodynamic, hydraulic, or hydrological models (Hsiao et al., 2021; Santiago-Collazo et al., 2021; Ikeuchi et al., 2017). The coupling could be one-way, two-way, or dynamic coupling. Another approach is deep learning and data fusion (Muñoz et al., 2021), and data assimilation (Muñoz et al., 2022).

**Material and methods:** *This section is of major concern since the methodology presented in this study is not technically sound. I agree that developed countries might face challenges to implement hydrodynamic models due to data scarcity and computational resource limitations. Nevertheless, assuming that those countries develop such a model and want to implement the proposed scheme. Are 6000 simulations really necessary to train machine learning algorithms? Would not it be better to wisely sample a small set of realistic forcing conditions that effectively lead to compound flooding? Although machine learning is a 'data-hungry' technique, I consider thousand of hydrodynamic simulations a bit exaggerated. I suggest the authors train the models with a small sample size (e.g., hundred of simulations) and report the results. This can help reduce computation time associated with hydrodynamic simulations. How long does it take to run the 2D-model of the Kapuas-Kecil in a regular desktop computer?*

***Response:***
We are sorry because there is a misunderstanding between what we did and what the reviewer thinks we did regarding the term "scenarios." Actually, we only ran ten months of hydrodynamic simulations, and then we extracted only 6000 pairs of data points (hourly water level in Pontianak vs. its associated input variables). To run the 2D model of the Kapuas-Kecil on a regular desktop computer (with 16 GB memory) took about 12 hours.

***Section 2.2:*** *There is no information regarding model calibration. This is critical as the authors rely on hydrodynamic simulations to train the machine learning algorithms. Referring to a pre-print/unpublished work (Sampurno et al., 2021) for additional details of the model is not acceptable. Please describe the model calibration process in detail.*
***Response:***
Actually, the previous work we mentioned has been accepted for publication and will be available online soon. However, the suggestion has been followed. We added a new section dedicated to model setup and calibration:

**2.3 Hydrodynamic model setup and calibration**
In order to run the hydrodynamic model, we defined a computational domain that covers both the river and the ocean parts. Next, we generated an unstructured mesh to cover the domain, with a resolution of 50 m over the riverbanks, 400 m over the coast near the river mouth, 1 km over the rest of the coastline, and 5 km over the offshore (Fig. 3). The multi-scale mesh was generated using an algorithm developed by Remacle and Lambrechts (2018). Next, we set the bathymetry constructed from two data sets: first, the river and estuary bathymetry maps, obtained from the Indonesian Navy (Kästner, 2019), and second, the Karimata Strait bathymetry, obtained from BATNAS (BATimetri NASional, 2021). Furthermore, we set the bulk bottom drag coefficients, which are $2.5\times10^{-3}$ over the ocean (which corresponds to a sandy seabed) and $1.9\times10^{-2}$ over the river bed (Kästner et al., 2018). Lastly, we imposed the rainfall, as observed by the Pontianak Maritime Meteorological Station (PMMS).

The hydrodynamic model simulation is forced by wind and atmospheric pressure from ECMWF (Hersbach et al., 2020), and tides from TPXO (Egbert and Erofeeva, 2002). As upstream boundary conditions, we imposed discharge from the Kapuas River and the Landak River. The discharge data were retrieved from the Global Flood Monitoring System (GFMS) (Wu et al., 2014).

We also imposed runoff, which was obtained by converting rainfall over the Kapuas Kecil River catchment area as an inlet water flux at some channels entering the domain. The runoff of every channel was calculated from rainfall data using SWAT+ (Bieger et al., 2017), which considered the pressure, the humidity, and other weather parameter input. Unfortunately, during the tuning of the SWAT model, the correlation between the output of the model (runoff) and the observation data is still low (0.32). However, we decided to use the output as the channels' inlet boundary condition in the hydrodynamic model because the channel runoff volume is much less than the river discharge. Therefore, we assumed that it does not significantly affect the hydrodynamics of the river.

To evaluate the SLIM model performance, we ran a simulation for January 2019 and compared the simulated water elevation with the observations in Pontianak. The model errors correspond to an NSE of 0.87 and an RMSE of 0.12 m (Fig. 4). This RMSE is deemed sufficiently small to consider model outputs as a good proxy of the real system (Moriasi et al., 2015).

We simulated the hydrodynamics with oceanic, atmospheric, and river forcings to forecast flood events based on the water levels. Based on the Pontianak Maritime Meteorological Station report, the city is flooded when the water level exceeds 2.5 m. We, therefore, set this value as the threshold

of a flood event. We ran the hydrodynamic model for ten months and extracted the output hourly to produce the scenarios (see Table 1). Then, we selected 6,000 sample points of the predicted water levels at Pontianak with their associated input dataset. We merged the data as a single dataset to train the machine learning model, encompassing all possible flood events resulting from the combination of the external forcings. The dataset shows that several flooding occurred within the simulations, indicated by sample points with water elevations greater than 2.5 m (Fig. 5).

*__Section 2.4__. Another point of major concern is the calibration of machine learning algorithms. It is no clear whether the authors tuned random forest and support vector machine in the training phase or not. In that regard, the training dataset (e.g., 6000 model outputs with the associated input variables) should have been split into training/validation datasets to conduct hyperparameter tuning and so prevent overfitting issues. Using all model outputs to train the algorithms (as reported here) and relying on default parameter-values is not a wise use of machine learning (e.g., Random requires tuning of the number of trees, sample leaf, sample split, etc.). The authors should conduct a thorough 'hyperparameter' tuning as it substantially improves the performance of machine learning algorithms.*

*__Response:__*
The suggestion has been followed. We did the calibration for both algorithms. We searched for the optimal value of the number of variables randomly sampled as candidates at each split for Random Forest and got 16 as the optimal one. For the SVM algorithm, we tuned it to select the correct choice of kernel parameters, which is crucial for obtaining good results. We tested four kernel algorithms, i.e., linear, polynomial, radial basis, and sigmoid. We found that the radial basis kernel gave the best performance for the SVM algorithm.

We added these sentences on section 2.4.2, for RF:

To obtain the optimal parameter for the RF, we first tune the algorithm by searching for the optimal value of the number of variables randomly sampled as candidates at each split (mtry). As a result, the optimal number is 16 (Fig. 7).

[Figure]

**Figure:** Tuned randomForest algorithm for the optimal number of variables randomly sampled as candidates at each split (mtry) parameter.

And we added this paragraph for MLR:

To obtain the best performance of the MLR algorithm, we did a statistical analysis to evaluate the multicollinearity among the predictor variables using the Variance Inflation Ratio (VIF). Since multicollinearity negatively affects the performance of the MLR model, VIF can help reduce the number of predictors (Alipour et al., 2020). Here, we found that some variables have VIF more significant than 5, which indicates a potentially severe correlation between these variables in the model (Fig. 8). Therefore, combined with the output of *MI* analysis, we removed some variables which have low *MI* and high VIF.

[Figure]

**Figure:** Variance Inflation Factor values of all predictor's variables in 3 months of observational data.

And this paragraph for SVM:

Since kernel function is critical in SVM, we tuned the SVM algorithm to obtain good results by selecting the most appropriate kernel parameter. We tested four kernels, i.e., linear, polynomial, radial basis, and sigmoid, as the candidates. We found that the radial basis kernel performed the best for the SVM algorithm.

***Results.*** *"Even though all algorithms perform very well during the training phase, the performances are different during the testing phases". This is known as overfitting (Ying, 2019) and occurs because random forest and support vector machine are not calibrated/tuned in the training phase.*

***Response:***
As mentioned above, we are following the suggestion. We did the tuning in the training phase for all algorithms to avoid overfitting.

**Technical corrections**

***L10*** *and thorough the text: There are odd terms that should be corrected like hydrodynamic modeling instead of 'water level modeling'.*
***Response:*** The suggestion has been followed.

*L28*: *What is the growth rate in the last decade?*
*Response:* In the Low Elevated Coastal Zone, the population will increase from 638 million in 2000 by 58% to 71% by 2050 (Merkens et al., 2016).

*L38*: *Please elaborate more on non-structural measures. This sentence is not clear.*
*Response:* The suggestion has been followed. We added more sentences in the paragraph:
Non-structural measures mean any actions to manage the risk of compound flooding without involving a physical construction (UNDRR, 2022), including land-use regulations, flood forecasting, warning systems, floodproofing and disaster prevention, and preparedness and response mechanisms.

*L40*: *Which issue? Please explain clearly.*
*Response:* We replaced word "the issue" with "the water-level prediction issue"

*L45-47*: *'Machine learning can enable us…' How? Please, elaborate more on this. More references are needed discussing the benefits of machine learning for water level prediction and/or flood forecasts.*
*Response:* The suggestion has been followed. We added more details after this sentence:
For instance, by assuming that flood events are stochastic, machine learning can predict major flood events based on certain probability distributions from the historical discharge data (Mosavi et al., 2018). In some cases, their performance is even more accurate than traditional statistical models (Xu and Li, 2002). In other words, we can prepare strategies to mitigate the flood risks using a machine learning model.

*L96*: *Please locate Pontianak in Figure 2.*
*Response:* Figure 2 updated. We added the perimeter of Pontianak on the map.

*L97*: *Section 2.3 is very short (4 lines) and should be included in the previous section.*
*Response:* The suggestion has been followed.

*L65*: *More details of the study area are needed. What is the catchment size, average river flow, tidal regime, rate of local sea level rise at the Kapuas River?*
*Response:* The suggestion has been followed. We added these sentences on the paragraph:
Its water catchment area spreads over about 93000 km2 (about 12.5% of the Borneo Island area, Fig. 1), with about 66.7% of it consisting of forests (Wahyu et al., 2010). Its topography comprises hills over its upstream, covered mainly by Acrisol soils (Fig. 2). In contrast, its downstream comprises plains with more heterogeneous soil types (Fig. 2), such as *Humic Gleysols* (derived from grass or forest vegetation) and *Dystric Fluvisols* (young soil in alluvial deposits).
And, create a new paragraph in this section:
As a tidal river, the tidal regime within the Kapuas River downstream area is mixed but mainly diurnal (Kästner, 2019). The dominant tidal constituent is K1, O1, P1, M2, and S2 (Pauta, 2018). The average tidal amplitude downstream is set in a microtidal regime, with a mean spring range of 1.45 m at its river mouth (Kästner, 2019).

***L115****: I suggest a more robust statistical analysis to evaluate multicollinearity among the variables (e.g., Variance Inflation Ratio (VIF), see Alipour (2020)). Multicollinearity negatively affects the performance of support vector machine and multilinear regression models. VIF can help reduce the number of predictors.*

***Response:*** The suggestion has been followed. We now test the multicollinearity among the predictor variables and found some variables having VIF greater than 5, which indicates a potentially severe correlation between these variables in the model. Therefore, we removed some variables based on this VIF values combine with MI coefficients, without reducing the model's performance.

***L155-156****: NSE and RMSE might improve after a thorough hyperparameter tuning of the machine learning algorithms.*

***Response:*** We did the tuning and reduced the number of predictors. The NSE and RMSE are better in the training and the testing phases but still not too different in the implementation phase.

***L158****: There are no such inundation scenarios (no flood maps). This should be clarified and better replaced for water level scenarios.*

***Response:***

The suggestion has been followed. We replace this sentence:

*We then simulated several inundation scenarios to produce datasets used to train the machine learning model.*

With the following:

We simulated the hydrodynamics with different oceanic, atmospheric, and river forcings to forecast flood events based on the water levels. Based on the Pontianak Maritime Meteorological Station report, the city is flooded when the water level exceeds 2.5 m. We, therefore, set this value as the threshold of a flood event. We ran the hydrodynamic model for ten months and extracted the output hourly to produce the scenarios (see Table 1).

We also moved this sentence from the result section to the Material and Method section, as suggested by Reviewer2.

***Table 1****. What are the criteria to come up with those range of values?*
***Response:***
We set those ranges based on minimum and maximum observation data from 2016 to 2021.

***Figure 2****. Scale bar and north arrow are missing.*
***Response:***
The figure has been updated. We added scale bare and north arrow. We also added grid-coordinates and combined it with the bathymetry map (as suggested by Reviewer 2). We changed Figure 2 to Figure 3.

***Figure 6.*** *Comparison of predicted and 'simulated' hourly water levels of training data. There are no observed water levels in the training phase.*
***Response:*** The suggestion has been followed. We updated the Figure.

***Figure 8.*** *X-axis is not observation but hydrodynamic simulation.*
***Response:*** The suggestion has been followed. We updated the Figure.

***L295.*** *There are references not included in the main text. See for example Rozum et al., 2020.*
***Response:*** Actually, there is no **Rozum et al., 2020**. It is still the part of **Hersbach, et al, 2020**. We will put more space between references to make them more easily checked.
***The full reference is:***
**Hersbach**, H., Bell, B., Berrisford, P., Hirahara, S., Horányi, A., Muñoz-Sabater, J., Nicolas, J., Peubey, C., Radu, R., Schepers, D., Simmons, A., Soci, C., Abdalla, S., Abellan, X., Balsamo, G., Bechtold, P., Biavati, G., Bidlot, J., Bonavita, M., Chiara, G., Dahlgren, P., Dee, D., Diamantakis, M., Dragani, R., Flemming, J., Forbes, R., Fuentes, M., Geer, A., Haimberger, L., Healy, S., Hogan, R. J., Hólm, E., Janisková, M., Keeley, S., Laloyaux, P., Lopez, P., Lupu, C., Radnoti, G., Rosnay, P., **Rozum**, I., Vamborg, F., Villaume, S., and Thépaut, J.: The ERA5 global reanalysis, Q. J. R. Meteorol. Soc., 146, 1999–2049, https://doi.org/10.1002/qj.3803, 2020

**Reference:**

[revised manuscript text omitted]

---

## Author Comment (AC2)

**Response to the second reviewers' comments (RC2) on the paper "*Integrated hydrodynamic and machine learning models for compound flooding prediction in a data-scarce estuarine delta*".**

We want to thank the reviewer for taking the time to review our paper. Their comments have been beneficial and helped us to improve the article. In what follows, the reviewer's comments are presented in italic type and our response in roman type.

**General Comments**

*I will like to congratulate the authors for writing a great manuscript. This manuscript was very well put together that tells a story in an organized and scientific manner. However, some essential details were omitted from this first draft. First, the hydrodynamic model calibration/validation is missing from section 2.2. I understand that the authors reference the reader to another article (currently under review) for more details on the hydrodynamic model. But in any flood modeling, it is crucial to discuss the hydrodynamic model calibration/validation, especially when the outputs of this model will be used as input in another model. At a minimum, the authors should dedicate a paragraph (if not a subsection) to discuss the results and method of the hydrodynamic model calibration/validation without going into much detail since the authors can reference another article.*

*Second, the flooding scenarios selected (e.g., Table 1) used to describe compound floods using the hydrodynamic model lacks information. For example, the authors should explain why they selected that certain combination of environmental factors and related to any observations or datasets. Finally, the authors do not give any information on the coupling occurring between the different hydrodynamic models to assess compound floods. This aspect is crucial in this type of research, and at least a subsection should be dedicated to explaining this model pass information between them to account for compound floods. Nevertheless, before I can accept the article for publishing, it needs to go through a major revision that will require a re-revision from the reviewers. The authors should recall that the main purpose of publishing a research article like this is to adopt the proposed methods and apply them to their region of interest. Therefore, it is crucial to include as much detail as pertinent to replicate the proposed work. Please find below some specific comments and questions that need to be addressed in the revised version of the author's manuscript.*

**Specific Comments**

*Section 1*

*• Line 25: the authors should include the following publications as part of this citation: Santiago-Collazo et al. (2021), Gori et al. (2022), Hsiao et al. (2021), Ghanbari et al. (2021)*

**Response:** The suggestion has been followed. We added the recommended citations. Here is the updated sentence:

Compound flooding in low-lying coastal areas is a recognized hazard that can be exacerbated by global warming (Hao and Singh, 2020; Santiago-Collazo et al., 2021; Gori et al., 2022; Hsiao et al., 2021; Ghanbari et al., 2021).

*• Line 26: the authors should include the following publications as part of this citation: Ikeuchi et al. (2017), Wahl et al. (2015)*

**Response:** The suggestion has been followed. We added the recommended citations. Here is the updated sentence:

Compound flooding hazard is derived from the interaction of storm surge penetration, riverine flooding, and intense rainfall over the areas (as the impact of extreme meteorological events) that coincide or nearly coincide (Bilskie and Hagen, 2018; Ikeuchi et al., 2017; Wahl et al., 2015).

*• It needs a paragraph of a literature review of previous modeling frameworks that uses a deterministic model to train an A.I. model. This will help put in context to the reader earlier attempts of this modeling approach. This might be the first attempt to simulate compound flood events, but other studies might focus on different processes such as subsurface flow and even at other disciplines such as transportation and structural engineering. Some questions that can be answered from including this paragraph might be the following:*

*o Is this the first study that uses a deterministic-A.I. modeling framework to estimate compound floods? If not, how was it then, and what was their approach?*

*o Have other researchers used a deterministic-A.I. modeling framework to estimate different parameters outside of surface flow physics?*

**Response:** We add a new paragraph in this section regarding the modeling frameworks to estimate compound floods, as follows:

"A new paradigm that combines deterministic and machine learning components has been proposed and implemented to tackle data and computational limitations in environmental modeling (Krasnopolsky and Fox-Rabinovitz, 2006; Goldstein and Coco, 2015). However, to the best of our knowledge, no previous modeling frameworks have developed a deterministic model to train a machine learning model for compound flooding studies. As a common practice, compound flood modeling typically uses the coupling of two or more hydrodynamic, hydraulic, or hydrological models (Hsiao et al., 2021; Santiago-Collazo et al., 2021; Ikeuchi et al., 2017). The coupling could be one-way, two-way, or dynamic coupling. Another approach is deep learning and data fusion (Muñoz et al., 2021), and data assimilation (Muñoz et al., 2022)."

**Section 2.1**

*• It will be beneficial for the reader to include an additional figure with the study area's topographic/bathymetric elevation map and land use/land cover maps and soil type maps since all these parameters will affect surface runoff modeling than subsequently will affect the compound flood magnitude. If there is no such data available as a map format, the authors should indicate it in the manuscript. This will highlight the date scarcity in the region.*

**Response:**

The suggested maps have been added as Figure 2:

[Figure]

Figure 2: Kapuas water catchment area (upper left), Digital elevation map (upper right) retrieved from SRTM (Farr et al., 2007), Land cover maps (lower left) retrieved from CGLOPS1 (Buchhorn et al., 2020), and Soil type maps (lower right) retrieved from FAO (Sanchez et al., 2009) for the Kapuas River catchment area.

- *Consider adding the Kapuas River watershed area and compare it with the total island extent. This will help the reader put the extension of this watershed into context, rather than just saying that it is the longest island river.*

**Response:**

We add this sentence to the paragraph:

The Kapuas River is the longest inland river in Indonesia (Goltenboth et al., 2006). The basin is located in the western part of Borneo Island (Fig. 1). The water catchment area spreads over about 93,000 km$^2$ (about 12.5% of the Borneo Island area, Fig. 1), with about 66.7% of it consisting of forests (Wahyu et al., 2010). The upstream topography comprises hills covered mainly by Acrisol soils (Fig. 2), and the downstream consists of plains with more heterogeneous soil types (Fig. 2), such as *Humic Gleysols* (derived from grass or forest vegetation) and *Dystric Fluvisols* (young soil in alluvial deposits). The river is vital for the local communities as a source of fresh water and a transportation system.

- *Figure 1: need to include in the figure caption that the solid black line represents the Kapuas River Watershed on the insert map. Also, mention that the blue lines represent waterbodies.*

**Response:**

We modified the figure 1 caption as follows:

**Figure 1.** The region of interest (ROI), where the green enclosed perimeter represents the city of Pontianak. The solid black line represents the Kapuas River Watershed in the inset map, and the blue lines represent waterbodies.

**Section 2.2**

• *Line 76: need to add a reference to cite the SLIM 2d hydrodynamic model. Similar to the SWAT+ citation on Line 94*

**Response:**

Reference has been added in the text:

To simulate hydrodynamics within the Kapuas River delta, we use the multi-scale hydrodynamic model SLIM 2D (Lambrechts et al., 2008; Gourgue et al., 2009; Remacle and Lambrechts, 2016).

• *Need to add a paragraph or subsection of the calibration/validation of both deterministic models used: SLIM 2D and SWAT+*

**Response:**

We added a new subsection:

**"2.3 Hydrodynamic model setup and calibration**

In order to run the hydrodynamic model, we defined a computational domain that covers both the river and the ocean parts. Next, we generated an unstructured mesh to cover the domain, with a resolution of 50 m over the riverbanks, 400 m over the coast near the river mouth, 1 km over the rest of the coastline, and 5 km over the offshore (Fig. 3). The multi-scale mesh was generated using an algorithm developed by Remacle and Lambrechts (2018). Next, we set the bathymetry constructed from two data sets: first, the river and estuary bathymetry maps, obtained from the Indonesian Navy (Kästner, 2019), and second, the Karimata Strait bathymetry, obtained from BATNAS (BATimetri NASional, 2021). Furthermore, we set the bulk bottom drag coefficients, which are $2.5 \times 10^{-3}$ over the ocean (which corresponds to a sandy seabed) and $1.9 \times 10^{-2}$ over the river bed (Kästner et al., 2018). Lastly, we imposed the rainfall, as observed by the Pontianak Maritime Meteorological Station (PMMS).

The hydrodynamic model simulation is forced by wind and atmospheric pressure from ECMWF (Hersbach et al., 2020), and tides from TPXO (Egbert and Erofeeva, 2002). As upstream boundary conditions, we imposed discharge from the Kapuas River and the Landak River. The discharge data were retrieved from the Global Flood Monitoring System (GFMS) (Wu et al., 2014).

We also imposed runoff, which was obtained by converting rainfall over the Kapuas Kecil River catchment area as an inlet water flux at some channels entering the domain. The runoff of every channel was calculated from rainfall data using SWAT+ (Bieger et al., 2017), which considered the pressure, the humidity, and other weather parameter input. Unfortunately, during the tuning of the SWAT model, the correlation between the output of the model (runoff) and the observation data is still low (0.32). However, we decided to use the output as the channels' inlet boundary

condition in the hydrodynamic model because the channel runoff volume is much less than the river discharge. Therefore, we assumed that it does not significantly affect the hydrodynamics of the river.

To evaluate the SLIM model performance, we ran a simulation for January 2019 and compared the simulated water elevation with the observations in Pontianak. The model errors correspond to an NSE of 0.87 and an RMSE of 0.12 m (Fig. 4). This RMSE is deemed sufficiently small to consider model outputs as a good proxy of the real system (Moriasi et al., 2015).

We simulated the hydrodynamics with different oceanic, atmospheric, and river forcings to forecast flood events based on the water levels in Pontianak. Based on the Pontianak Maritime Meteorological Station report, the city is flooded when the water level exceeds 2.5 m. We, therefore, set this value as the threshold of a flood event. We ran the hydrodynamic model for ten months and extracted the output hourly to produce the scenarios (see Table 1). Then, we selected 6,000 sample points of the predicted water levels at Pontianak with their associated input dataset. We merged the data as a single dataset to train the machine learning model, encompassing all possible flood events resulting from the combination of the external forcings. The dataset shows that several flooding occurred within the simulations, indicated by sample points with water elevations greater than 2.5 m (Fig. 5). "

For SWAT model validation, since we don't have observation data for channels within the KRD, we used the observation data taken on 2012 upstream of the Landak River ($0.741^0$N, $110.101^0$E). Unfortunately, the Pearson correlation of the model is only 0.32 (still low correlated). However, the flows from the channels are much less than the flow in the river stream. Therefore, we decided to use the SWAT model outputs as the boundary condition for the SLIM model at the channels' inlets within the KRD. While we set the boundary condition at the river stream using discharge retrieved from the global flow model, GFMS (http://flood.umd.edu/).

[Figure]

Figure 4b: SWAT model output validation with respect to observational data in Landak upstream ($0.741^0$N, $110.101^0$E).

•        *Line 85-89: How far inland does the mesh extend through the river? Does it penetrate through the riverine floodplain or stop at the river bank's height? Does the digital elevation model (DEM) used in the hydrodynamic model (details are not given) penetrate beneath the water to capture the full river bathymetry (i.e., description of the terrain surface underwater), so the riverine*

*cross-section is described fully, or does it reflect the water surface elevation? If the complete riverine cross-section is not available from observed data, which cross-sectional area do the authors use? These details are not given in the text nor Figure 2.*

**Response:**

Our mesh extended inland 70 km to cover the delta. The mesh stops at the river banks. Here, our bathymetry covers both the ocean and the river bathymetry. We added this information in the new subsection, **2. 3 Hydrodynamic model setup and calibration,** in this paragraph:

In order to run the hydrodynamic model, we defined a computational domain that covers both the river and the ocean parts. Next, we generated an unstructured mesh to cover the domain, with a resolution of 50 m over the riverbanks, 400 m over the coast near the river mouth, 1 km over the rest of the coastline, and 5 km over the offshore (Fig. 3). The multi-scale mesh was generated using an algorithm developed by Remacle and Lambrechts (2018). Next, we set the bathymetry constructed from two data sets: first, the river and estuary bathymetry maps, obtained from the Indonesian Navy (Kästner, 2019), and second, the Karimata Strait bathymetry, obtained from BATNAS (BATimetri NASional, 2021). Furthermore, we set the bulk bottom drag coefficients, which are $2.5 \times 10^{-3}$ over the ocean (which corresponds to a sandy seabed) and $1.9 \times 10^{-2}$ over the river bed (Kästner et al., 2018). Lastly, we imposed the rainfall, as observed by the Pontianak Maritime Meteorological Station (PMMS).

• *Figure 2: include a bathymetry elevation as a color-filled contour with the unstructured mesh, so the reader can examine if there are any canyons or through underwater that will affect the coastal processes flood modeling. The authors may also consider adding the mesh resolution like a color map, see Figure 3 on Bislkie et al. (2020).*

**Response:**

The suggestion has been followed. We combine the bathymetry map and the mesh in Figure 2 as Figure 3. Since the bathymetry is already in color, we keep the mesh elements in a single color (black).

[Figure]

**Figure 3:** The hydrodynamic model domain is discretized with an unstructured mesh whose resolution is set to 50 m along the riverbanks, 400 m along the coast near the estuary, 1 km over the rest of the coastline, and 5 km offshore. The bathymetry of the model domain ranges from ~100 m depth offshore to 1 m in the river mouth.

• *Line 90-92: information about the different environmental factors considered in the study was given in Table 1. However, information regarding the astronomical tide forcing is not given, just from the model that was obtained. I think that more information should be given since, at the discussion session, the authors concluded that tidal forcing is the factor that most affects the compound flood levels in the regions. The authors should answer the following questions within the text:*
o *What is the average tidal amplitude (e.g., micro-tidal, meso-tidal, macro-tidal)?*
o *What is the dominant tidal constituent (e.g., M2, S1, K1, etc.)?*
o *What is the tidal regime (i.e., period) at the region (e.g., diurnal, semi-diurnal, or mixed)?*
**Response:**
The suggestion has been followed. We added this paragraph in the study area sub-section:
"As a tidal river, the tidal regime within the Kapuas River delta is mixed but mainly diurnal (Kästner, 2019). The dominant tidal constituent is K1, O1, P1, M2, and S2 (Pauta, 2018). The average tidal amplitude within the delta is set in a microtidal regime, with a mean spring range of 1.45 m at its river mouth (Kästner, 2019)."

*• Line 92-93: the authors should explain in more detail the coupling procedure between SWAT+ and SLIM 2D. Also, the authors should locate on a map the riverine boundary conditions in the SLIM 2D model and clearly specify the total amount of locations. The following questions should be answered in the text of the manuscript:*

*o What type of coupling is occurring between the models (e.g., one-way, two-way, tightly, or fully coupling)?*

*o How often (e.g., each computational time step) does the exchange of information happen?*

*o Do the SWAT+ model runs first and independently, and once it finishes the simulation, it passes the information to SLIM 2D, or do both models run simultaneously?*

*o Is the location of the riverine boundary conditions in the SLIM 2D model inland enough (i.e., away from the coast) that coastal processes will not affect the water levels? If not, the authors should justify the selection of that location.*

**Response:**

Here, we considered a one-way coupling, where the SWAT+ model runs first and independently. The SWAT+ model only produces the flow through channels that enter the river stream within the KRD. Then, we used these channel outlets as boundary conditions for the SLIM model. At the same time, the river discharges (boundary conditions) for the main streams were retrieved from GFMS (http://flood.umd.edu/) at about 70 km and 40 km from the river mouth. Since the GFMS calculates the flow using Integrated Multi-Satellite Retrievals for GPM (IMERG) precipitation information as input, the coastal processes do not affect the model output (predicted river flow).

*• Line 93-95: the authors do not give any information regarding the hydrologic modeling using SWAT+. Since it does not reference another publication, at least a subsection should be dedicated to providing more details of this model. This information is crucial since the SWAT+ model computes the pluvial and fluvial processes in the compound flood simulation in SLIM 2D. For example, Silva-Araya et al. (2018) described their hydrologic and hydrodynamic model in separate subsections before describing the coupling technique in an additional subsection. The following questions should be answered in the text of the manuscript: o Does infiltration processes are taken into consideration?*

*o How many sub-watershed was the Kapuas River watershed divided into so it was suitable to model in SWAT+?*

*o What is the extent of the SWAT+ model? A figure should be included.*

*o What was the temporal resolution of this model?*

*o Did the rainfall vary in time and space through the domain?*

**Response:**

[Figure]

The Kapuas River watershed is divided into 14 sub-basins. However, since we retrieved the discharges of the Kapuas River from GFMS at the middle stream, we consider only two sub-basins for the SWAT model (yellow area in the figure). The SWAT model temporal resolution is daily. Since both sub-basins have a wide area, the rainfall varies in time and space throughout the entire domain.

*• Line 95-96: the authors should include in Figure 1 (or on an additional figure) the location of the gauge where the observational data was obtained. What type of observational data was used to evaluate the model performance (e.g., stage, discharge, high-water marks, etc.)?*
**Response:**
The location of the gauge has been added to Figure 1. The type of data observation is hourly water levels.

**Section 2.3**
*This section lacks much essential information for the reader, and it is not clear. This section is one of the most important in the manuscript since it will control the compound flood event being simulated. The following questions should be answered in the text of the manuscript:*
*o Table 1:*
    *▪From where were these values chosen, and why these values themselves?*
    *▪Why do the tables display only a single value of discharge, whereas, in Line 109, the authors said that the datasets (including riverine variables) were recorded hourly? Is the value shown in the table represents the annual peak discharge, the average value, etc.?*

*o Why 6,000 simulations and not 1,000 or 10,000? Need to justify the author's decision.*
*o How was the combination of the different parameters chosen? Did the authors use any statistical approaches, such as a Monte Carlo Simulation, or used a random distribution?*
*o Why the hydrodynamic model was run for 10 months and not 12 or 6?*
**Response:**

The range of parameters is based on the minimum and maximum observational data measured by PMMS. Regarding river discharges, we set them to vary in some simulations (data points number 4200 to 5400) while they are kept constant in the others. The range of parameters is based on the minimum and maximum observational data measured by PMMS. We chose 6000 because we only ran the model for ten-month. Then, we extracted from each month 600 sample points. The combination of the dependent and predictors parameters was chosen randomly. We ran the hydrodynamic model for only ten months because we assumed that all possible compound flooding scenarios had been represented within this duration.

**Section 2.4.1**
• *Line 108-109: why did the authors select just one and two hours before the flood event as the prior conditions? It has been shown that rainfall events that occur three days before a flood event have measurable effects on the compound flood levels (Bilskie et al., 2021). The authors need to justify their selection. What was the SLIM output temporal resolution?*
**Response:**

We chose one and two hours before the event because we assumed that the region of interest (i.e., the city of Pontianak) is located only about 20 km from the river mouth. The river width between these two locations is wide (300 m within the city to 900 m close to the river mouth) without dykes, dams, or any hydraulic structure. So, the flow along this river downstream can move freely. Therefore, based on the previous events observed by PMMS, the inundation as the impact of the interaction between these factors will happen in a short time.
The SLIM output temporal resolution is hourly.

• *Table 2: the biggest tidal variations occur within 6 to 12 hours before/after their peak level, depending on the tidal regime. Therefore, it does not make sense to vary their tidal elevation (which is not given in Table 1 nor the text) by one or two hours since the values are very similar. It will make more sense that the authors tested scenarios that considered the high and low tidal elevation, which can be 6 to 12 hours apart.*
**Response:**

The suggestion has been followed. We evaluated the mutual information between water level dynamics and prior tidal elevation of one to 12 hours (see the figure bellows):

[Figure]

The figure shows that the MI coefficients of 6 and 12 hours apart of tidal elevations (Tide6 and Tide12) are lower than one and two hours prior. Therefore, we only chose the one and two hours of tidal elevations prior and omitted the others as the predictor.

**Section 3**

• *Line 155-157: need to cite other studies that confirm your statement that a model with those values of NSE and RMSE is a "good proxy" of the real system.*

**Response:** The suggestion has been followed. We modified the sentences and moved it to sub-section 2.3:

"To evaluate the SLIM model performance, we ran a simulation for January 2019 and compared the simulated water elevation with the observations in Pontianak. The model errors correspond to an NSE of 0.87 and an RMSE of 0.12 m (Fig. 4). This RMSE is deemed sufficiently small to consider model outputs as a good proxy of the real system (Moriasi et al., 2015). "

• *Line 158-161: this can be moved to Section 2.*
**Response:** The suggestion has been followed.

• *Figure 5: the authors should comment if the low impact of rainfall to compound flood events might be related to the small amount of rainfall used. Also, can the selection of a lumped-parameter hydrologic model (SWAT+) used in this study affect the surface runoff quantity fed into the hydrodynamic model?*

**Response:**

The low impact of rainfall on compound flood events might be related to the huge transport of the river discharge compared to the run-off which enters the river stream within the domain. Since we define a flooding event due to the height of the river water level passing the threshold, the rainfall impact on compound flooding is categorized as low. However, based on the observation, a single excessive rainfall is enough to trigger urban flooding within the city for a few hours, while the water level within the river is low. This urban flooding could be due to the poor quality of the urban drainage system. However, this specific phenomenon still cannot directly be captured by the water level observation located within the river. Consequently, the increase in the river water level due to the heavy rain is not well represented by the model and becomes the limitation of this study.

*• Add a vertical axis label to Figure 5.*

**Response:** The suggestion has been followed.

*• Improve the resolution of Figures 3, 6, and 7.*
**Response:** The suggestion has been followed.

**Section 4**
*• The authors should comment if the low accuracy of the A.I. model during the testing phase is related to the calibration/validation of the hydrodynamic model? If the hydrodynamic model is inaccurate in predicting real-life floods, then the A.I. model will have low accuracy.*
**Response:**

The suggestion has been followed. We add this sentences in this section:

"However, the integrated model proposed in this study also has some limitations. Firstly, the accuracy of the machine learning model built depends on the accuracy of the hydrodynamic model. The more accurate the hydrodynamic model in predicting observational floods, the better the machine learning model will perform. Therefore, we need to tune the hydrodynamic model as accurately as possible."

*• Why is the biggest impact of the compound flood levels due to tidal conditions? How do these findings relate to the physical processes occurring at this location? Have other studies drawn similar conclusions regarding the importance of tides in a compound flood event? The authors should talk more about this.*
**Response:**

The tidal conditions have the most significant impact on compound flooding hazard in Pontianak because the city is just 20 km away from the river mouth. The city and the river mouth area are connected with a wide river stream (about 300 m within the city to 900 m close to the river mouth) without dykes, dams, or any hydraulic structure. Therefore, the tide still strongly dominates the hydrodynamics of the river within the city, and the compound flooding tends to happen there at high tide. We have not yet found the other study with a similar conclusion, unfortunately.

**Section 5**
• *The conclusion session needs improvement. For example, topics in the discussion section should be at the conclusion section, such as modeling limitations and future research.*
**Response:**
The suggestion has been followed. We updated this section as follows:
This study shows that an integrated approach between the hydrodynamic and the machine learning models successfully overcomes modeling river water-level and predicting compound flooding hazards in a data-scarce environment with limited computational resources. Therefore, the approach is suitable for local water management agencies in developing countries that are faced with these issues. However, the accuracy of the machine learning model depends on the accuracy of the hydrodynamic model. If the hydrodynamic model is inaccurate in predicting real-life floods, the machine learning model's accuracy will also be lower. Besides, it has not yet optimally captured the urban flooding due to excessive rainfall. Considering more indicators representing this kind of flooding is essential to enhance the model's capability in the future. Regarding the implementation in Pontianak, we found that the machine learning model with the RF algorithm has the most accurate output compared to the other algorithms. In addition, the tidal elevation, measured one hour prior, is the main predictor for water level modeling in the study area.

• *The authors should also include as part of their modeling limitation the use of the SWAT+ model to quantify the pluvial and fluvial processes in their compound flood event. The SWAT+ model is a conceptual-based, lumped-parameter hydrologic model. Therefore, this model has many limitations when computation spatially- and time-varying surface flow compared to physically-based, distributed-parameter hydrologic models capable of having a spatial distribution of precipitation and watershed properties through a computational grid.*
**Response:**
The suggestion has been followed. We added some sentences as follows in the section 2.3:
"The runoff of every channel was calculated from rainfall data using SWAT+ (Bieger et al., 2017), which considered the pressure, the humidity, and other weather parameter input. Unfortunately, during the tuning of the SWAT model, the correlation between the output of the model (runoff) and the observation data is still low (0.32). However, we decided to use the output as the channels' inlet boundary condition in the hydrodynamic model because the channel runoff volume is much less than the river discharge. Therefore, we assumed that it does not significantly affect the hydrodynamics of the river."

**Technical Corrections**
• *Line 111: it should say "Statistic tool" and not "atistic tool"*
*Response:* We add this sentence to replace the typo:

[revised manuscript text omitted]

---

## Referee Report (RR1)

**General comments**

The authors should include a limitation section/subsection highlighting all the potential errors that the authors faced when developing this model so that other research may build upon this. Currently, the model's limitations are all over the manuscript, but it will benefit the reader if they are all summarized in a section. For example, the following are some limitations collected through the manuscript

- The authors assumed that channel runoff volume would not affect the hydrodynamics of the river due to its small volume compared to the riverine volume.
- The authors assume that within 10 months, all the possible compound flood scenarios occurred.
- Lack of drainage system in the hydrodynamic model for the urban region of the domain.
- The accuracy of the machine learning model depends on the accuracy of the hydrodynamic model.

**Specific Comments**

"A new paradigm that combines deterministic and machine learning components has been proposed and implemented to tackle data and computational limitations in environmental modeling (Krasnopolsky and Fox-Rabinovitz, 2006; Goldstein and Coco, 2015). However, to the best of our knowledge, no previous modeling frameworks have developed a deterministic model to train a machine learning model for compound flooding studies. As a common practice, compound flood modeling typically uses the coupling of two or more hydrodynamic, hydraulic, or hydrological models (Hsiao et al., 2021; Santiago-Collazo et al., 2021; Ikeuchi et al., 2017). The coupling could be one-way, two-way, or dynamic coupling. Another approach is deep learning and data fusion (Muñoz et al., 2021), and data assimilation (Muñoz et al., 2022)."

What type of environmental modeling did the authors refer to? There are many examples, such as subsurface flow, pollutant transport, etc. Please give examples within the same sentence.

2.3 Hydrodynamic model setup and calibration

In how many locations did the authors impose the runoff in the model? It only refers to "some channels entering the domain", but it is important to specify the amount as a minimum and preferably show their locations on a map.

The value of 0.32 given in parenthesis when talking about the correlation between the SWAT model and the observation is unclear. Is this 0.32 referring to r-squared value, RMSE, etc.? Be more specific in the manuscript text.

The location of the upstream riverine boundary condition is missing details. For example, how far away was this boundary from the coast? In any compound flood model, the upstream riverine boundary condition should be inland enough that any water level variation due to tides is negligible. Thus, only riverine forces are the ones driving the flow downstream. The authors should mention in the manuscript text that the distance from these locations' coasts is minimal.

The authors claim that the channel runoff volume is much less than the river discharge. How much less is it? It needs to be quantified in the manuscript.

RMSE and NSE acronyms are not defined before their appearance in this new subsection. I strongly recommend the authors create a small subsection describing the performance criteria used to verify the model. In this way, the authors could properly define each metric (e.g., NSE, R2, RMSE), including the equations used.

The authors mentioned that they sampled 6,000 points from their predicted water levels but did not mention how it was sampled. Did the authors use a random sampling technique or a probabilistic distribution to select these 6,000 points? Please specify this in the manuscript text.

Add the proper citation to reference the GFMS rather than the URL in the text.

Response to comments on Line 92-93

The authors should include the response to the reviewer in the manuscript text with respect to the coupling details between SWAT and SLIM

Response to comments on Line 93-95

The authors should include the figure in response to the reviewer on the manuscript since it gives a better perspective of the hydrologic processes in the study area,

---

## Author Response (AR2)

**Response to the second round comments of the second reviewer (RC2) on the paper:**
***"Integrated hydrodynamic and machine learning models for compound flooding prediction in a data-scarce estuarine delta"***

We want to thank again the reviewer2 for taking the time to re-review our paper. Their comment has been beneficial and helped us to improve the article. In what follows, the reviewer's comments are presented in italic type and my response in roman type.

**General Comments**

*The authors should include a limitation section/subsection highlighting all the potential errors that the authors faced when developing this model so that other research may build upon this. Currently, the model's limitations are all over the manuscript, but it will benefit the reader if they are all summarized in a section. For example, the following are some limitations collected through the manuscript*

*• The authors assumed that channel runoff volume would not affect the hydrodynamics of the river due to its small volume compared to the riverine volume.*
*• The authors assume that within 10 months, all the possible compound flood scenarios occurred.*
*• Lack of drainage system in the hydrodynamic model for the urban region of the domain.*
*• The accuracy of the machine learning model depends on the accuracy of the hydrodynamic model.*

**Response:**
The suggestion has been followed. We added the following subsection in the manuscript:

**2.6. Model limitations**

During the development process, we encountered potential errors that could be highlighted as model limitations. Firstly, we assumed that channel runoff volume would not affect the hydrodynamics of the river due to its small volume compared to the riverine volume. The average daily discharge of the Kapuas River and the Landak River during the simulation is about 4,137 m3/s and 406 m3/s. At the same time, the total daily runoff of all channels which enter the hydrodynamic model domain in the KRD is about 32 $m^3$/s. The runoff contributes only about 0.7% of the total inlets in the hydrodynamic simulations; therefore, we assumed it is insignificant.

Secondly, we assumed that all the possible compound flood scenarios would occur within ten months. Since we already set some extreme values in the predictor parameters during the time, we assumed that all possible causes that drive compound flooding in the domain are represented. However, this assumption may not be accurate.

Next, we only imposed the runoffs as inlets on the river banks in the hydrodynamic model domain. Hence, the model did not capture the hydrodynamic processes in the channels within the city. It means that the inundation processes in Pontianak were still not well represented. The model still lacks drainage systems for the urban region.

Moreover, the accuracy of the machine learning model depends on the hydrodynamic model's accuracy. The more accurate the hydrodynamic model in predicting observational floods, the better the machine learning model will perform. Therefore, we need to tune the hydrodynamic model as accurately as possible.

Furthermore, since the rainfall impact on river water level is minor compared to other parameters, the model could not optimally capture urban flooding due to excessive rainfall. Based on the field observation, the city is shortly inundated if rain falls excessively for a few hours. This inundation could be due to the poor quality of the urban drainage system. Unfortunately, this phenomenon is not directly captured by the water level observation located within the river. The increase in the river water level due to the heavy rain is minor.

Lastly, the model relies on the predicted input parameters such as weather parameters and river discharges to predict the future water level. Consequently, the more biased the predictors, the higher the uncertainty in the water-level prediction. Therefore, observational data as input parameters are needed to reduce the uncertainty and create a more robust model. "

**Specific Comments**
*"A new paradigm that combines deterministic and machine learning components has been proposed and implemented to tackle data and computational limitations in environmental modeling (Krasnopolsky and Fox-Rabinovitz, 2006; Goldstein and Coco, 2015). However, to the best of our knowledge, no previous modeling frameworks have developed a deterministic model to train a machine learning model for compound flooding studies. As a common practice, compound flood modeling typically uses the coupling of two or more hydrodynamic, hydraulic, or hydrological models (Hsiao et al., 2021; Santiago-Collazo et al., 2021; Ikeuchi et al., 2017). The coupling could be one-way, two-way, or dynamic coupling. Another approach is deep learning and data fusion (Muñoz et al., 2021), and data assimilation (Muñoz et al., 2022)."*

*What type of environmental modeling did the authors refer to? There are many examples, such as subsurface flow, pollutant transport, etc. Please give examples within the same sentence.*
**Response:**
The suggestion has been followed. We edited and added examples within the sentence become:
"A new paradigm that combines deterministic and machine learning components has been proposed to tackle data and computational limitations in environmental modeling, such as hybrid climate models (Krasnopolsky and Fox-Rabinovitz, 2006) and an ML model for 2D surface water catchment problems (Maxwell et al., 2021)."

**2.3 Hydrodynamic model setup and calibration**
*In how many locations did the authors impose the runoff in the model? It only refers to "some channels entering the domain", but it is important to specify the amount as a minimum and preferably show their locations on a map.*
**Response:**
The suggestion has been followed. We showed the locations in Figure 4 (Just added to the manuscript). Then, we added the information in the sentence.
"We also imposed runoff, obtained by converting rainfall over the Kapuas Kecil River catchment area as an inlet water flux at 15 channels entering the domain (Fig. 4)."

*The value of 0.32 given in parenthesis when talking about the correlation between the SWAT model and the observation is unclear. Is this 0.32 referring to r-squared value, RMSE, etc.? Be more specific in the manuscript text.*

**Response:**

The value of 0.32 referred to the Pearson correlation coefficient (*r*). Therefore, we edited the sentence become:

"Unfortunately, during the tuning of the SWAT+ model, the correlation between the model's output (runoff) and the observation data is still low (Pearson correlation coefficient = 0.32)."

*The location of the upstream riverine boundary condition is missing details. For example, how far away was this boundary from the coast? In any compound flood model, the upstream riverine boundary condition should be inland enough that any water level variation due to tides is negligible. Thus, only riverine forces are the ones driving the flow downstream. The authors should mention in the manuscript text that the distance from these locations' coasts is minimal.*

**Response:**

The suggestion has been followed. We added detailed information about discharges by inserting the below sentences (blue text) in the second paragraph of subsection 2.3:

"The hydrodynamic model simulation is forced by wind and atmospheric pressure from ECMWF (Hersbach et al., 2020), and tides from TPXO (Egbert and Erofeeva, 2002). As upstream boundary conditions, we imposed discharge from the Kapuas River and the Landak River. The discharge data were retrieved from the Global Flood Monitoring System (GFMS) (Wu et al., 2014) at about 70 km and 40 km from the river mouth (Fig. 4). Since the GFMS calculates the flow using Integrated Multi-Satellite Retrievals for GPM (IMERG) precipitation information as input, the coastal processes do not affect the model output (predicted river flow)."

*The authors claim that the channel runoff volume is much less than the river discharge. How much less is it? It needs to be quantified in the manuscript.*

**Response:**

The suggestion has been followed. We described the comparison between total daily runoffs and discharges in the new subsection (2.6 Model limitations) as follows:

"The average daily discharge of the Kapuas River and the Landak River during the simulation is about 4,137 m3/s and 406 m3/s. At the same time, the total daily runoff of all channels which enter the hydrodynamic model domain in the KRD is about 32 m3/s. The runoff contributes only about 0.7% of the total inlets in the hydrodynamic simulations; therefore, we assumed it is insignificant."

*RMSE and NSE acronyms are not defined before their appearance in this new subsection. I strongly recommend the authors create a small subsection describing the performance criteria used to verify the model. In this way, the authors could properly define each metric (e.g., NSE, R2, RMSE), including the equations used.*

**Response:**

The suggestion has been followed. We created a new subsection to describe the performance criteria which used (2.3. Metrics for model performance evaluation).

*The authors mentioned that they sampled 6,000 points from their predicted water levels but did not mention how it was sampled. Did the authors use a random sampling technique or a probabilistic distribution to select these 6,000 points? Please specify this in the manuscript text.*
**Response:**
We use a random sampling technique to obtain the sampled points. We added this additional information within the associated sentence in the manuscript (blue text):
"Then, we selected 6,000 sample points of the predicted water levels at Pontianak with their associated input dataset using a random sampling technique."

*Add the proper citation to reference the GFMS rather than the URL in the text.*
**Response:**
The suggestion has been followed. We added citation for GFMS.

**Response to comments on Line 92-93**
*The authors should include the response to the reviewer in the manuscript text with respect to the coupling details between SWAT and SLIM*
**Response:**
The suggestion has been followed. We inserted the below sentences (blue text) in the paragraph:

"We also imposed runoff, obtained by converting rainfall over the Kapuas Kecil River catchment area as an inlet water flux at some channels entering the domain. The runoff of every channel was calculated from rainfall data using SWAT+ (Bieger et al., 2017), which considered the pressure, the humidity, and other weather parameter input. Here, we use one-way coupling, where the SWAT+ model runs first and independently. The SWAT+ model only produces the flow of channels that enter the river stream within the KRD. Then, we used these channel outlets as boundary conditions for the SLIM model. Unfortunately, during the tuning of the SWAT model, the correlation between the model's output (runoff) and the observation data is still low (NSE = 0.32). However, we decided to use the output as the channels' inlet boundary condition in the hydrodynamic model because the channel runoff volume is much less than the river discharge. Therefore, we assumed that it does not significantly affect the hydrodynamics of the river."

**Response to comments on Line 93-95**
*The authors should include the figure in response to the reviewer on the manuscript since it gives a better perspective of the hydrologic processes in the study area.*
**Response:**
The suggestion has been followed. We included the below figure in the manuscript.

[Figure]

**Figure 4:** The Kapuas River watershed and its sub-basins. Since the discharges of the Kapuas River are retrieved at the middle stream, only two sub-basins are considered for the SWAT+ model (yellow area). The runoffs (channel outlets of the SWAT+ model that enter the river stream within the KRD) are set as inlets for the hydrodynamic model domain.

**References**

Bieger, K., Arnold, J. G., Rathjens, H., White, M. J., Bosch, D. D., Allen, P. M., Volk, M., and Srinivasan, R.: Introduction to SWAT+, a Completely Restructured Version of the Soil and Water Assessment Tool, J. Am. Water Resour. Assoc., 53, 115–130, https://doi.org/10.1111/1752-1688.12482, 2017.

Egbert, G. D. and Erofeeva, S. Y.: Efficient inverse modeling of barotropic ocean tides, J. Atmos. Ocean. Technol., 19, 183–204, 2002.

Hersbach, H., Bell, B., Berrisford, P., Hirahara, S., Horányi, A., Muñoz-Sabater, J., Nicolas, J., Peubey, C., Radu, R., Schepers, D., Simmons, A., Soci, C., Abdalla, S., Abellan, X., Balsamo, G., Bechtold, P., Biavati, G., Bidlot, J., Bonavita, M., Chiara, G., Dahlgren, P., Dee, D., Diamantakis, M., Dragani, R., Flemming, J., Forbes, R., Fuentes, M., Geer, A., Haimberger, L., Healy, S., Hogan, R. J., Hólm, E., Janisková, M., Keeley, S., Laloyaux, P., Lopez, P., Lupu, C., Radnoti, G., Rosnay, P., Rozum, I., Vamborg, F., Villaume, S., and Thépaut, J.: The ERA5 global reanalysis, Q. J. R. Meteorol. Soc., 146, 1999–2049, https://doi.org/10.1002/qj.3803, 2020.

Krasnopolsky, V. M. and Fox-Rabinovitz, M. S.: A new synergetic paradigm in environmental numerical modeling: Hybrid models combining deterministic and machine learning components, Ecol. Modell., 191, 5–18, https://doi.org/10.1016/J.ECOLMODEL.2005.08.009, 2006.

Maxwell, R. M., Condon, L. E., and Melchior, P.: A Physics-Informed, Machine Learning Emulator of a 2D Surface Water Model: What Temporal Networks and Simulation-Based Inference Can Help Us Learn about Hydrologic Processes, Water 2021, Vol. 13, Page 3633, 13, 3633, https://doi.org/10.3390/W13243633, 2021.

Wu, H., Adler, R. F., Tian, Y., Huffman, G. J., Li, H., and Wang, J.: Real-time global flood estimation using satellite-based precipitation and a coupled land surface and routing model, Water Resour. Res., 50, 2693–2717, https://doi.org/10.1002/2013WR014710, 2014.